# SqueezeLLM: Dense and Sparse Quantization

## Abstract

Generative Large Language Models (LLMs) have demonstrated remarkable results for a wide range of tasks. However, deploying these models for inference has been a significant challenge due to their unprecedented resource requirements. This has forced existing deployment frameworks to use multi-GPU inference pipelines, which are often complex and costly, or to use smaller and less performant models. In this work, we demonstrate that the main bottleneck for generative inference with LLMs is memory bandwidth, rather than compute, specifically for single batch inference. While quantization has emerged as a promising solution by representing weights with reduced precision, previous efforts have often resulted in notable performance degradation. To address this, we introduce SqueezeLLM, a post-training quantization framework that not only enables lossless compression to ultra-low precisions of up to 3-bit, but also achieves higher quantization performance under the same memory constraint. Our framework incorporates two novel ideas: (i) *sensitivity-based non-uniform quantization*, which searches for the optimal bit precision assignment based on second-order information; and (ii) the *Dense-and-Sparse decomposition* that stores outliers and sensitive weight values in an efficient sparse format. When applied to the LLaMA models, our 3-bit quantization significantly reduces the perplexity gap from the FP16 baseline by up to $2.1\times$ as compared to the state-of-the-art methods with the same memory requirement. Furthermore, when deployed on an A6000 GPU, our quantized models achieve up to $2.3\times$ speedup compared to the baseline.

## 1 Introduction

Recent advances in Large Language Models (LLMs) trained on massive text corpora, with up to hundreds of billions of parameters, have showcased their remarkable problem-solving capabilities across various domains Brown et al. (2020); Raffel et al. (2020); Scao et al. (2022); Du et al. (2022); Hoffmann et al. (2022); Chowdhery et al. (2022); Smith et al. (2022); Zhang et al. (2022); Thoppilan et al. (2022); Touvron et al. (2023a). However, deploying these models for inference has been a significant challenge due to their demanding resource requirements. For instance, the LLaMA-65B Touvron et al. (2021) model requires at least 130GB of RAM to deploy in FP16, which exceeds current GPU capacity. Even storing such large-sized models has become costly and complex.

As will be discussed in Sec. 3, the main performance bottleneck in LLM inference for generative tasks is memory bandwidth rather than compute. This means that the speed at which we can load and store parameters becomes the primary latency bottleneck for memory-bound problems, rather than arithmetic computations. However, recent advancements in memory bandwidth technology have been significantly slow compared to the improvements in computes, leading to the phenomenon known as the Memory Wall Patterson (2004). Consequently, researchers have turned their attention to exploring algorithmic methods to overcome this challenge.

One promising approach is quantization, where model parameters are stored at lower precision, instead of the typical 16 or 32-bit precision used for training. For instance, it has been demonstrated that LLM models can be stored in 8-bit precision without performance degradation Yao et al. (2022), where 8-bit quantization not only improves the storage requirements by half but also has the potential to improve inference latency and throughput. As a result, there has been significant research interest in quantizing models to even lower precisions. A pioneering approach is GPTQ Frantar et al. (2022) which uses a training-free quantization technique that achieves near-lossless 4-bit quantization for large LLM models with over tens of billions of parameters. However, achieving high quantization performance remains challenging, particularly with lower bit precision and for relatively smaller models (e.g., $< 50B$ parameters) such as the recent LLaMA Touvron et al. (2023a).

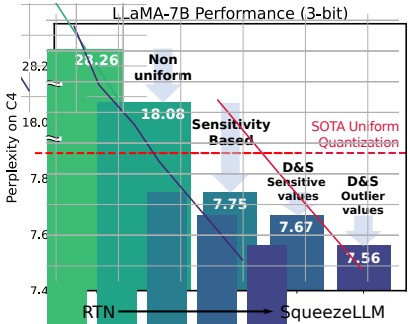
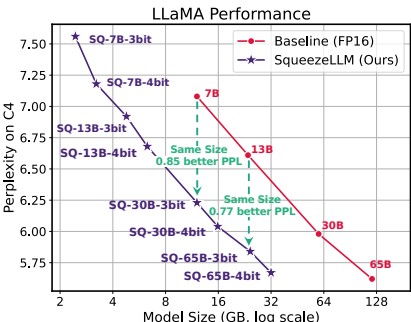

**Figure 1:** *(Left) SqueezeLLM incorporates two key approaches: (i) sensitivity-based non-uniform quantization (Sec. 4.1), where quantization bins are allocated closer to sensitive values, and (ii) the Dense-and-Sparse decomposition (Sec. 4.2), which retains both sensitive values and outlier values as full-precision sparse format. When applied to LLaMA-7B with 3-bit quantization, our method outperforms the state-of-the-art methods Frantar et al. (2022); Lin et al. (2023) by a large perplexity margin of over 0.3 on the C4 benchmark. (Right) By applying our methods to LLaMA models of varying sizes, we can achieve improved trade-offs between perplexity and model size.*

In this paper, we conduct an extensive study of low-bit precision quantization and identify limitations in existing approaches. Building upon these insights, we propose a novel solution that achieves lossless compression and improved quantization performance even at precisions as low as 3 bits.

**Contributions.** We first present performance modeling results demonstrating that *the memory, rather than the compute, is the primary bottleneck* in LLM inference with generative tasks (Sec. 3 and Fig. A.1). Building on this insight, we introduce SqueezeLLM, a post-training quantization framework with a novel *sensitivity-based non-uniform quantization* and *Dense-and-Sparse decomposition*. These techniques enable ultra-low-bit precision with reduced model sizes and faster inference without compromising model performance. Our detailed contributions include:

- **Sensitivity-based Non-Uniform Quantization:** We demonstrate that uniform quantization, as commonly adopted in prior works, is sub-optimal for LLM inference for two reasons. First, the weight distributions in LLMs exhibit clear non-uniform patterns (Fig. 2). Second, the inference computation in prior works does not benefit from uniform quantization as the arithmetic is performed in FP16 precision, not in reduced precision. To address these, we propose a novel sensitivity-based non-uniform quantization method to achieve a more optimal quantization scheme for LLMs. Our approach significantly improves the perplexity of the LLaMA-7B model at 3-bit precision from 28.26 of uniform quantization to 7.75 on the C4 dataset (Sec. 4.1).

- **Dense-and-Sparse Quantization:** We observe that weights in LLMs contain significant outliers, making low-bit quantization extremely challenging. To address this, we propose a simple solution that decomposes weights into dense and sparse components. The sparse part holds outlier values in full precision using efficient sparse storage methods and leverages an efficient sparse kernel for minimal inference overhead. This allows the dense part to have a more compact range (up to $10\times$) and aids quantization. By extracting only 0.45% of the weight values as the sparse component, we further improve the perplexity of LLaMA-7B from 7.75 to 7.58 on the C4 dataset (Sec. 4.2).

- **Evaluation:** We extensively test SqueezeLLM on various models on language modeling tasks using the C4 and WikiText2 benchmarks, where we find that SqueezeLLM consistently outperforms existing quantization methods by a large margin across different bit precisions (Sec. 5.2). We also demonstrate the potential of SqueezeLLM in quantizing instruction following models by applying it to the Vicuna models Chiang et al. (2023) on the MMLU benchmark Hendrycks et al. (2021) and the Vicuna benchmark Chiang et al. (2023) (Sec. 5.3). Furthermore, our deployed models on A6000 GPUs also exhibit significant gains in latency of up to $2.4\times$ compared to the FP16 baseline, showcasing the effectiveness of our method in terms of both quantization performance and inference efficiency (Sec. 5.4).

## 2 RELATED WORK

In Sec. A.1, we offer an overview and related works of Transformer quantization, with a particular emphasis on Post-Training Quantization (PTQ) and non-uniform quantization, which are the primary focus of our work. Among the various challenges in low-bit Transformer quantization, one

key issue is the presence of outliers Kovaleva et al. (2021), which can unnecessarily increase the quantization range. To address this issue, outlier-aware quantization methods have been investigated Bondarenko et al. (2021); Dettmers et al.; Wei et al. (2022; 2023). Notably, Dettmers et al. keeps outlier activations in floating-point, while Wei et al. (2022) transfers outlier factors to later layers without affecting functionality. These focus on activations, which is not a concern in our work where all activations are maintained in floating-point. Our Dense-and-Sparse quantization instead tackles *weight* outliers for low-bit LLM quantization.

Concurrently to our work, SpQR Dettmers et al. (2023) also explores a method for extracting outliers in the context of quantization. However, SpQR employs a different sensitivity metric based on the Optimal Brain Surgeon (OBS) framework Hassibi et al. (1993); Hassibi & Stork (1993), where the weights are quantized in a way that the output activations of each layer are not perturbed. In contrast, our approach is based on the Optimal Brain Damage (OBD) framework LeCun et al. (1990) where the weights are quantized to preserve the final output of the model. While both approaches show promise, we have observed that the OBD method yields better quantization performance since it is a direct measure of the end-to-end performance degradation after quantization (Sec. A.5.4).

More importantly, SqueezeLLM avoids techniques that can introduce high overhead and complexity when implementing lossless quantization. First, SqueezeLLM does not incorporate grouping. Our Dense-and-Sparse scheme provides a *direct* solution to prevent outlier values from negatively impacting quantization performance, eliminating the need for using the grouping strategy as an indirect solution (Sec. A.5.3). In contrast, SpQR requires fine-grained grouping (e.g., group size 16) which increases the model size and complicates the quantization pipeline by necessitating the bi-level quantization scheme. Second, the sensitivity-based non-uniform quantization in SqueezeLLM allows for much smaller (e.g., 0.05%) or even zero sparsity levels to achieve accurate quantization. This is crucial for reducing the model size as well as inference speed since higher sparsity levels can degrade inference latency. By avoiding grouping and utilizing smaller or zero sparsity levels, SqueezeLLM achieves accurate and fast quantization while pushing the average bit precision down to 3-bit, all while employing a simpler quantization pipeline and implementation.

Another concurrent work is AWQ Lin et al. (2023) which improves the weight-only quantization scheme for LLMs by introducing scaling factors to reduce the quantization error of a few important weights. However, their approach is also based on the OBS framework, where sensitivity is determined by the magnitude of activations. In Sec. 5, we demonstrate that our method consistently outperforms AWQ in terms of quantization performance across various models and applications.

## 3 MEMORY WALL

Inference behavior broadly falls into two categories: *compute-bound* inference that is limited by computational throughput, and *memory-bound* inference that is bottlenecked by the rate at which data can be fed into the processing cores from memory. *Arithmetic intensity*, the ratio of compute to memory operations, is a typical metric used to assess this behavior. High and low arithmetic intensity indicates a compute-bound and memory-bound problem, respectively. For memory-bound problems, the speedup can be achieved by reducing the memory traffic rather than compute since the compute units in hardware are often underutilized waiting to receive data from memory.

Generative LLM inference exhibits extremely low arithmetic intensity compared to other workloads[1] Kim et al. (2023). This is because it consists almost entirely of matrix-vector operations, which limits the achievable data reuse as each weight load can only process a single vector for a single token, and cannot be amortized across the multiple vectors for different tokens. This low arithmetic intensity needs to be contrasted with the compute operations on a typical GPU which is orders of magnitude higher than the memory operations[2]. The disparity between compute and memory bandwidth, along with the growing memory requirements of deep learning, has been termed the *Memory Wall* problem Gholami et al. (2021b). To further illustrate this problem in generative LLMs, we used a simple roofline-based performance modeling approach Kim et al. (2023) to study LLaMA-7B's runtime on an A5000 GPU with different bit precisions (Fig. A.1). Here, we assume that all computations are kept at FP16. Despite this, we can clearly see that the latency decreases linearly as we reduce the bit precision, indicating that the main bottleneck is memory, not compute.

---

[1]To be precise, we limit this discussion to single batch inference where the arithmetic involves matrix-vector operations. For large batch inference or different model architectures, compute can become important.

[2]For instance, A5000 GPU has peak computational throughput of 222 TeraFLOPs per second, which is $290\times$ higher than the peak memory bandwidth of 768 GigaBytes per second.

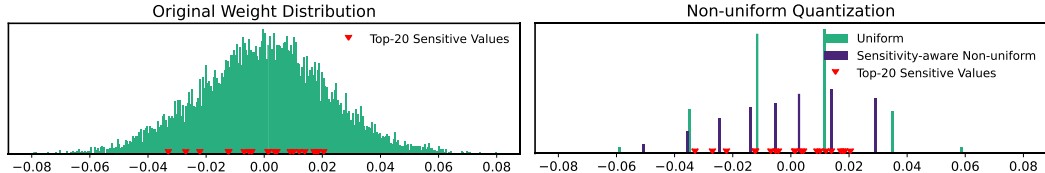

**Figure 2:** *(Left) The weight distribution of one output channel in LLaMA-7B. The top-20 most sensitive values are marked in red. (Right) Weight distributions after 3-bit quantization using uniform and sensitivity-based non-uniform quantization. In the latter case, the quantized values are more clustered around the sensitive values.*

In summary, in generative LLM inference, loading weight matrices into memory is the primary bottleneck, while the cost of dequantization and FP16 computation is relatively insignificant. Thus, by quantizing just the weights to lower precision, while leaving the activations in full precision, we can attain significant speedup as well as reduced model size. Given this insight, the appropriate strategy is to *minimize the memory size even if it may add overhead to arithmetic operations*.

## 4 METHODOLOGY

### 4.1 SENSITIVITY-BASED NON-UNIFORM QUANTIZATION

In Fig. 2 (Left), we plot an exemplary weight distribution in LLaMA-7B that demonstrates a non-uniform pattern. The main task for quantization is to find an optimal way to allocate distinct quantized values (e.g., 8 for 3 bits) in a way that preserves model performance. As discussed, a widely used approach in the recent LLM quantization works Frantar et al. (2022); Dettmers et al. (2023); Lin et al. (2023) is uniform quantization where the weight range is evenly divided into bins, and each bin is represented by a single integer number. This has two main issues. First, uniformly distributing quantized values is sub-optimal as weight distributions are typically non-uniform. Second, while the main advantage of uniform quantization is efficient integer computation, this does not lead to end-to-end latency improvement in memory-bound LLM inference. Therefore, we have chosen non-uniform quantization, which allows for a more flexible allocation of the representative values.

Finding an optimal non-uniform quantization configuration translates into solving a k-means problem. Given a weight distribution, the goal is to determine $k$ centroids that best represent the values (e.g., $k$=8 for 3-bit). This optimization problem for non-uniform quantization can be formulated as

$$Q(w)^* = \arg\min_Q \|W - W_Q\|_2^2, \tag{1}$$

where $W$ denotes the weights and $W_Q$ is the corresponding quantized weights (i.e., $[Q(w)$ for $w \in W]$), represented by $k$ distinct values $\{q_1, \cdots, q_k\}$. Here, the optimal solution $Q(w)^*$ can be obtained by 1-dimensional k-means clustering, which clusters the parameters into $k$ clusters and assign the centroid of each cluster as $q_j$'s. While this already outperforms uniform quantization, we propose an improved *sensitivity-based* clustering algorithm.

**Sensitivity-Based K-means Clustering.** The quantization objective is to represent the model weights with low-bit precision with minimal perturbation in the model output Dong et al. (2019). While quantization introduces perturbations in each layer, we need to minimize the overall perturbation with respect to the *final loss term*, rather than focusing on individual layers, as it provides a more direct measure of the end-to-end performance degradation after quantization LeCun et al. (1990). To achieve this, we need to place the k-means centroids closer to the values that are more sensitive with respect to the final loss, rather than treating all weight values equally as in Eq. 1. To determine more sensitive values, we perform Taylor expansion to analyze how the loss changes in response to perturbations in the weights $W$:

$$\mathcal{L}(W_Q) \simeq \mathcal{L}(W) - g^\top(W - W_Q) + \frac{1}{2}(W - W_Q)^\top H(W - W_Q) \tag{2}$$

where $g$ is the gradient and $H = \mathbb{E}[\frac{\partial^2}{\partial W^2}\mathcal{L}(W)]$ is the Hessian of the loss at $W$. Assuming that the model has converged to a local minimum, the gradient $g$ can be approximated as zero which gives us the following formula for computing how much the model gets perturbed after quantization:

$$Q(w)^* = \arg\min_Q (W - W_Q)^\top H(W - W_Q). \tag{3}$$

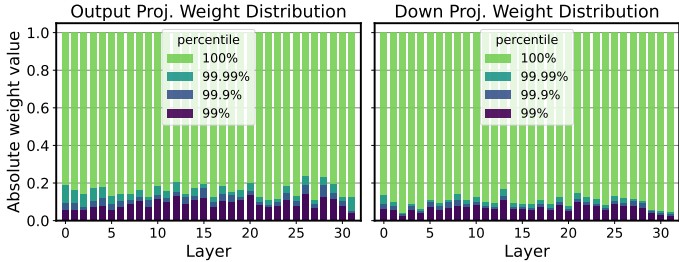

**Figure 3:** *The distributions of the (normalized) absolute weight values, for the output layers in MHA and the down layers in FFN across different layers in LLaMA-7B. Note that the distributions exhibit outlier patterns across all layers, with 99% of the values clustered within ~10% of the entire range.*

In the new optimization target, as compared to Eq. 1, the perturbation of each weight after quantization, i.e., $W - W_Q$, is weighted by the scaling factor introduced by the second-order derivative, $H$. This highlights the importance of minimizing perturbations for weights that have large Hessian values, as they have a greater impact on the overall perturbation of the final output. In other words, the second-order derivative serves as a measure of importance for each weight value.

Due to the cost of computing the Hessian, we use an approximation to the Hessian based on the Fisher information matrix $\mathcal{F}$, which can be calculated over a sample dataset $D$ as $H \simeq \mathcal{F} = \frac{1}{|D|} \sum_{d \in D} g_d g_d^\top$. This only requires computing gradient for a set of samples, which can be calculated efficiently with existing frameworks. To make the optimization objective in Eq. 3 more feasible, we further approximate the Fisher information matrix as a diagonal matrix by assuming that the cross-weight interactions are negligible. This simplifies our objective target as follows:

$$Q(w)^* \simeq \underset{Q}{\arg\min}(W - W_Q)^\top \mathrm{diag}(\mathcal{F})(W - W_Q) = \underset{Q}{\arg\min} \sum_{i=1}^{N} \mathcal{F}_{ii}(w_i - Q(w_i))^2. \quad (4)$$

An important consequence of Eq. 4 is the *weighted* k-means clustering setting, where the centroids will be pulled closer to these sensitive weight values. In Fig. 2, we provide an exemplary weight distribution of LLaMA-7B with the top 20 sensitive values based on the Fisher information. At the right, the quantized values assigned by uniform quantization (green) are compared to those assigned by the sensitivity-based k-means approach (purple), which achieves a better trade-off by placing centroids near sensitive values, effectively minimizing quantization error. With 3-bit LLaMA-7B, sensitivity-based non-uniform quantization achieves a much lower perplexity of 7.75 compared to the 28.26 perplexity of round-to-nearest uniform quantization on C4 (Fig. 1 and Sec. 5.2)

### 4.2 DENSE-AND-SPARSE QUANTIZATION

Another challenge in low-bit LLM quantization is outlier values Bondarenko et al. (2021); Dettmers et al.; Wei et al. (2022; 2023). In Fig 3, we plot the normalized weight distributions of different layers in LLaMA-7B, which demonstrate that ~99.9% of the weights are concentrated in a narrow range of ~10% of the entire distribution. Naively quantizing the weights with such large range, will significantly degrade performance, especially at low precisions such as 3-bits. However, the observation in Fig. 3 also implies opportunity. The range of the weight values can be contracted by a factor of $10\times$ simply by removing a small number of outlier values (e.g., 0.1%), yielding a significant improvement in quantization resolution. This will then help the sensitivity-based k-means centroids to focus more on the sensitive values rather than a few outliers.

Motivated by this, we introduce a method to filter out outliers from the weight matrix $W$ by performing a simple yet effective decomposition into a sparse matrix ($S$) containing the outliers and the remaining dense matrix ($D$) that can be quantized much more effectively thanks to its significantly reduced range of values. That is, $W = D + S$ where $D = W[T_{\min} \leq w \leq T_{\max}]$ and $S = W[w < T_{\min} \text{ or } w > T_{\max}]$. Here, $T_{\min/\max}$ are thresholds that define outliers based on the percentile of the distribution.

Importantly, the overhead of this decomposition is minimal, since the number of outlier values is small. Even in the most aggressive quantization experiments, we did not find it necessary to use $> 0.5\%$ of sparsity. Therefore, the sparse matrix can be stored efficiently using methods like compressed sparse row (CSR) format. Inference is also straightforward with the decomposition as

in $WX = DX + SX$, two kernels for dense and sparse multiplication can be overlapped, and the sparse part ($SX$) can benefit from sparse kernels (Sec. 4.3).

**Sensitivity-Based Sparse Matrix.** Beyond extracting outliers as a sparse matrix, we have also found it beneficial to extract a few highly sensitive values in weight matrices to make sure those values are represented exactly without any error. These values can be easily identified based on the Fisher information (Sec. 4.1). This offers two benefits. First, maintaining these sensitive values with FP16 precision minimizes their impact on the final output. Second, it prevents the centroids of Eq.4 from skewing towards the sensitive values, thus enhancing quantization for less sensitive weights. We have observed that extracting only 0.05% of these sensitive values across layers substantially enhances quantization performance (Sec. A.5). Altogether, with 3-bit LLaMA-7B, extracting 0.45% of outlier and sensitive values further reduces the perplexity from 7.67 to 7.56 (Fig. 1 and Sec. 5.2).

## 4.3 DENSE-AND-SPARSE KERNEL IMPLEMENTATION

While a natural question to consider is the impact of both the non-uniform and Dense-and-Sparse quantization on latency, we find it straightforward to implement them efficiently. We implement 3/4-bit LUT-based kernels for matrix-vector multiplication between compressed weight matrices and uncompressed activation vectors. These kernels load the compressed weights and dequantize them piece-by-piece to minimize memory bandwidth utilization. The compressed matrices store 3/4-bit indices, which correspond to LUT entries containing FP16 values associated with the bins obtained from non-uniform quantization. After dequantization, all arithmetic is performed in FP16.

To efficiently process our Dense-and-Sparse representation, we also develop CUDA kernels for sparse matrix-vector multiplication that load a matrix in CSR format and a dense activation vector, inspired by Evtushenko (2019). Since the non-zero entry distributions are highly skewed across rows (Sec. A.4), assigning a single thread per row can be inefficient due to the unbalanced amount of work assigned to different threads. Thus, we implement *balanced hybrid kernels* based on Flegar & Quintana-Ortí (2017) by assigning an equal number of nonzeros per thread; this leads to additional synchronization across threads since one row may be divided across several threads, but leads to a balanced work assignment. We set the number of threads such that there were 10 nonzero values assigned to each thread. The dense non-uniform kernel and balanced sparse kernels are launched in one call to avoid overhead from summing the output vectors from these separate operations.

## 5 EVALUATIONS

### 5.1 EXPERIMENT SETUP

In this section, we describe our experiment setup. More details can be found in Sec. A.3.

**Models and Datasets.** We have conducted comprehensive evaluations of SqueezeLLM using various models on different tasks. First, in the language modeling evaluation, we apply SqueezeLLM to LLaMA, LLaMA2 Touvron et al. (2023a;b), and OPT Zhang et al. (2022) on the C4 Raffel et al. (2020) and WikiText2 Merity et al. (2016) datasets. We also evaluate the domain-specific knowledge and problem-solving ability through zero-shot MMLU Hendrycks et al. (2021) using the Vicuna (v1.1 and v1.3) models. Finally, we evaluate the instruction following ability using the methodology in Chiang et al. (2023) that uses GPT-4 as a judge.

**Baseline Methods.** We compare SqueezeLLM against PTQ methods for LLMs including RTN, GPTQ Frantar et al. (2022), AWQ Lin et al. (2023) and SpQR Dettmers et al. (2023). To ensure a fair comparison, we use GPTQ *with* activation ordering throughout all experiments unless specified, which addresses the significant performance drop that would otherwise occur.

**Quantization Details.** For SqueezeLLM, we adopt channel-wise quantization where each output channel is assigned a separate lookup table. We use 2 different sparsity levels: 0% (dense-only) and 0.45% (0.05% sensitive values and 0.4% outlier values as discussed in Sec. 4.2). For measuring sensitivity, we use 100 random samples from the Vicuna training set for Vicuna models and C4 training set for the others. While grouping can also be incorporated with our method, we found it sub-optimal as compared to extracting sensitive/outlier values with sparsity (Sec. A.5.3).

**Latency Profiling.** We measure the latency and peak memory usage for generating 128 and 1024 tokens on an A6000 machine using the Torch CUDA profiler. As an official implementation of GPTQ (in particular, the grouped version) is not available, we implement an optimized kernel for single-batch inference based on the most active open-source codebase ( GPTQ-For-LLaMA).

**Table 1:** *Perplexity comparison of LLaMA models quantized into 3 and 4 bits using different methods including RTN, GPTQ, AWQ and SpQR on C4 and WikiText-2. We compare the performance of GPTQ, AWQ, and SqueezeLLM in groups based on similar model sizes. In the first group, we compare dense-only SqueezeLLM with non-grouped GPTQ. In the second group, we compare SqueezeLLM with a sparsity level of 0.45% to GPTQ and AWQ with a group size of 128. We add speedup and peak memory usage numbers for comparison. Further results for LLaMA-30/65B can be found in Tab. A.4. Detailed latency evaluation can be found in Tab. 3.*

| LLaMA-7B | 3-bit | | | | | 4-bit | | | | |
|---|---|---|---|---|---|---|---|---|---|---|
| **Method** | **Avg. Bits** (comp. rate) | **PPL** (↓) C4 | Wiki | **Speedup** (↑) | **Mem.** (GB, ↓) | **Avg. Bits** (comp. rate) | **PPL** (↓) C4 | Wiki | **Speedup** (↑) | **Mem.** (GB, ↓) |
| Baseline | 16 | 7.08 | 5.68 | 1× | 12.7 | 16 | 7.08 | 5.68 | 1× | 12.7 |
| RTN | 3 (5.33) | 28.26 | 25.61 | 2.3× | 2.9 | 4 (4.00) | 7.73 | 6.29 | 2.0× | 3.7 |
| GPTQ | 3 (5.33) | 9.55 | 7.55 | 2.3× | 2.9 | 4 (4.00) | 7.43 | 5.94 | 2.0× | 3.7 |
| SpQR | - | - | - | - | - | 3.94 (4.06) | 7.28 | 5.87 | 1.2×[†] | N/A |
| SqueezeLLM | 3.02 (5.29) | **7.75** | **6.32** | 2.1× | 2.9 | 4.05 (3.95) | **7.21** | **5.79** | 1.8× | 3.8 |
| GPTQ (g128, no reorder)[‡] | 3.24 (4.93) | 10.09 | 8.85 | 2.0× | 3.0 | 4.24 (3.77) | 7.80 | 6.07 | 1.6× | 3.8 |
| GPTQ (g128)[‡] | 3.24 (4.93) | 7.89 | 6.27 | 0.2× | 3.0 | 4.24 (3.77) | 7.21 | 5.78 | 0.4× | 3.8 |
| AWQ (g128) | 3.24 (4.93) | 7.90 | 6.44 | 2.0× | 3.0 | 4.24 (3.77) | 7.22 | 5.82 | 1.6× | 3.8 |
| SqueezeLLM (0.45%) | 3.24 (4.93) | **7.56** | **6.13** | 1.9× | 3.1 | 4.27 (3.75) | **7.18** | **5.77** | 1.7× | 4.0 |

| LLaMA-13B | 3-bit | | | | | 4-bit | | | | |
|---|---|---|---|---|---|---|---|---|---|---|
| **Method** | **Avg. Bits** (comp. rate) | **PPL** (↓) C4 | Wiki | **Speedup** (↑) | **Mem.** (GB, ↓) | **Avg. Bits** (comp. rate) | **PPL** (↓) C4 | Wiki | **Speedup** (↑) | **Mem.** (GB, ↓) |
| Baseline | 16 | 6.61 | 5.09 | 1× | 24.6 | 16 | 6.61 | 5.09 | 1× | 24.6 |
| RTN | 3 (5.33) | 13.24 | 11.78 | 2.7× | 5.3 | 4 (4.00) | 6.99 | 5.53 | 2.3× | 6.8 |
| GPTQ | 3 (5.33) | 8.22 | 6.22 | 2.7× | 5.3 | 4 (4.00) | 6.84 | 5.29 | 2.3× | 6.8 |
| SpQR | - | - | - | - | - | 3.96 (4.04) | 6.72 | 5.22 | 1.2×[†] | N/A |
| SqueezeLLM | 3.02 (5.30) | **7.08** | **5.60** | 2.4× | 5.4 | 4.04 (3.96) | **6.71** | **5.18** | 2.0× | 6.9 |
| GPTQ (g128, no reorder)[‡] | 3.25 (4.92) | 7.16 | 5.53 | 2.2× | 5.7 | 4.25 (3.77) | 6.71 | 5.18 | 1.9× | 7.2 |
| GPTQ (g128)[‡] | 3.25 (4.92) | 7.12 | 5.47 | 0.2× | 5.6 | 4.25 (3.77) | 6.70 | 5.17 | 0.4× | 7.0 |
| AWQ (g128) | 3.25 (4.92) | 7.08 | 5.52 | 2.2× | 5.7 | 4.25 (3.77) | 6.70 | 5.21 | 1.9× | 7.2 |
| SqueezeLLM (0.45%) | 3.24 (4.94) | **6.92** | **5.45** | 2.2× | 5.8 | 4.26 (3.76) | **6.68** | **5.17** | 1.9× | 7.3 |

## 5.2 MAIN RESULTS

Table 1 shows quantization results for LLaMA along with comparison with RTN, GPTQ and AWQ. The models are grouped based on their average bitwidth (i.e., model size) for a better comparison of size-perplexity trade-offs. See Fig. 4 for a visual illustration. Below we use LLaMA-7B as the main example for the discussions for the impact of dense-only and Dense-and-Sparse quantization, and subsequently discuss how these trends extend to larger models. We provide the full evaluation result on all LLaMA models in Tab. A.4.

**Dense-only Quantization.** In Tab. 1 (Top), we compare dense-only SqueezeLLM with 0% sparsity level and GPTQ without grouping. With 4-bit quantization, our method exhibits minimal degradation compared to the FP16 baseline, with only ∼0.1 perplexity degradation on C4 and WikiText2, while reducing the model size by 3.95×. Moreover, when compared to non-grouped GPTQ our method shows significant perplexity improvement of up to 0.22.

The performance gap between the two methods becomes more pronounced with 3-bit quantization. SqueezeLLM outperforms GPTQ by a substantial margin of 1.80/1.22 points on C4/WikiText2 with a 5.29× compression rate. This is only 0.67/0.55 points off from the FP16 baseline. This demonstrates the effectiveness of the sensitivity-based non-uniform method for ultra-low-bit quantization.

**Dense-and-Sparse Quantization.** By leveraging the Dense-and-Sparse quantization, we achieve a further reduction in the perplexity gap between the FP16 baseline and quantized models, as shown in Tab. 1. This improvement is particularly significant with 3-bit quantization, where extracting just 0.45% of the values yields around 0.2 perplexity improvement. This enables nearly lossless compression with less than 0.1/0.5 perplexity deviation from the FP16 baseline for 4/3-bit, respectively.

---

[†] Since SpQR does not release their kernel implementation, we conduct our best-effort comparison using their reported speedup numbers. See Sec. A.3 for details.

[‡] GPTQ with activation ordering incurs a significant latency penalty as elements in the same channel are associated with different scaling factors, resulting in distributed memory accesses (Sec. 5.4). While GPTQ *without* activation ordering alleviates the latency issue, comes at the cost of a substantial perplexity degradation.

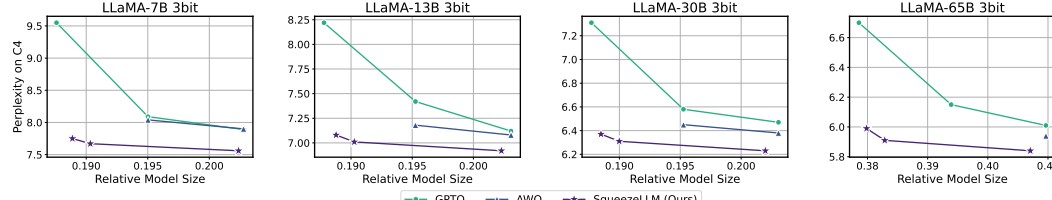

**Figure 4:** *Perplexity comparison PTQ methods for 3-bit LLaMA quantization, evaluated on C4. The x-axes are the relative model sizes with respect to the model size in FP16. Different size-perplexity trade-offs are achieved by adjusting the group size for GPTQ and AWQ and the sparsity level for ours. Our quantization method consistently and significantly outperforms GPTQ and AWQ across all model size regimes, with a more pronounced gap in lower-bit and smaller model sizes.*

**Table 2:** *Comparison of PTQ methods on zero-shot MMLU accuracy applied to Vicuna v1.1. We add speedup and peak memory usage for comparison.*

| Method | Avg. bit | 7B | | | 13B | | |
|---|---|---|---|---|---|---|---|
| | | Acc (↑) | Speedup (↑) | Mem (GB, ↓) | Acc (↑) | Speedup (↑) | Mem (GB, ↓) |
| Baseline | 16 | 39.1% | 1× | 12.7 | 41.2% | 1× | 24.6 |
| AWQ (g128) | 4.25 | 38.0% | 1.6× | 3.8 | 40.4% | 1.9× | 7.2 |
| SqLLM | 4.05 | 38.8% | 1.8× | 3.8 | 39.2% | 2.0× | 6.9 |
| SqLLM (0.45%) | 4.26 | **39.4%** | 1.7× | 4.0 | **41.0%** | 1.9× | 7.3 |
| AWQ (g128) | 3.25 | 36.5% | 2.0× | 3.0 | 37.6% | 2.2× | 5.7 |
| SqLLM | 3.02 | 36.0% | 2.1× | 2.9 | 37.2% | 2.4× | 5.4 |
| SqLLM (0.45%) | 3.24 | **37.7%** | 1.9× | 3.1 | **39.4%** | 2.2× | 5.8 |

Both GPTQ and AWQ use a grouping strategy to enhance performance with a slight overhead in model size. However, we demonstrate that SqueezeLLM with a sparsity level of 0.45% consistently outperforms both GPTQ and AWQ with a group size of 128 in all scenarios with comparable model sizes. This is more pronounced for 3-bit quantization, where SqueezeLLM with a 0.45% sparsity level outperforms both GPTQ and AWQ with a group size of 128 by up to more than 0.3 perplexity.

**Results on Larger Models.** In Tab. 1 (13B) and Tab. A.4 (30/65B), we observe that the trend in LLaMA-7B extends to larger models, where SqueezeLLM consistently outperforms other PTQ methods across all model sizes and bit widths. Such a trend is also visually illustrated in Fig. 4 for 3-bit quantization across all model sizes. Notably, even the dense-only version of SqueezeLLM achieves perplexity comparable to the grouped GPTQ and AWQ. With sparsity, we achieve further perplexity improvements, reducing the gap from the FP16 baseline to less than 0.1/0.4 perplexity points for 4/3-bit quantization. Notably, with 3-bit quantization, our approach achieves up to a 2.1× reduction in perplexity gap from the FP16 baseline compared to existing methods. Further ablation studies on our design choices, including sensitivity metrics, sparsity levels, and grouping, are provided in Sec. A.5, and additional results on LLaMA2 and OPT are in Sec. A.7.1.

### 5.3 QUANTIZATION OF INSTRUCTION FOLLOWING MODELS

Instruction tuning has emerged as a method for improving the model's ability to respond to user commands. We explore the quantization of instruction-following models to demonstrate the benefits of SqueezeLLM in terms of accuracy preservation by applying it to the Vicuna models, and evaluating the performance with the following approaches.

**Zero-shot MMLU Evaluation.** We first compare the baseline and quantized model on the zero-shot multitask problem-solving benchmark of MMLU. The weighted accuracy across all tasks is provided in Tab. 2 for Vicuna v1.1, including its quantized models using AWQ and SqueezeLLM. As we can see, SqueezeLLM achieves higher accuracy for both Vicuna-7B and 13B as compared to AWQ and also preserves the FP16 baseline accuracy with 4-bit quantization. It is also noteworthy that the 4-bit Vicuna-13B of SqueezeLLM has 2× smaller memory footprint than the 7B FP16 model, while still achieving a 2% higher accuracy. Additional results on Vicuna v1.3 are provided in Sec A.7.2.

**Instruction-Following Ability.** Another approach for evaluating instruction-following ability is to ask GPT-4 to rank the generated responses which is the method used by Chiang et al. (2023). The results are shown in Fig. 5. SqueezeLLM without sparsity achieves near-perfect performance (i.e., 50/50 split) with 4-bit quantization for both Vicuna-7B and 13B, outperforming GPTQ with the same model size. In the case of 3-bit quantization, SqueezeLLM outperforms both GPTQ and AWQ

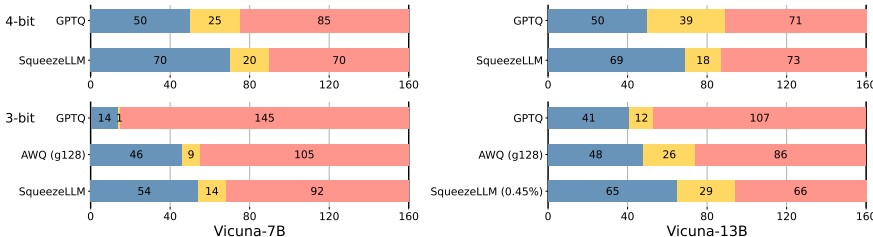

**Figure 5:** *Comparison of PTQ methods applied to Vicuna v1.1. Blue / yello / red represent the number of times that the quantized model won / tied / lost against the baseline FP16 model. This evaluation was performed using the methodology from Vicuna.*

with comparable model sizes. In the case of the Vicuna-13B model, achieving a near-perfect 50/50 split for 3-bit quantization.

## 5.4 HARDWARE DEPLOYMENT AND PROFILING

**Table 3:** *Latency (s) and peak memory usage (GB) of 3-bit LLaMA when generating 128 tokens on an A6000 GPU. The table compares the FP16 baseline, non-grouped and grouped GPTQ with activation ordering, and SqueezeLLM with different sparsity levels. For comparison, we include bitwidth and perpelxity on the C4 benchmark.*

| Method | Bit width | 7B | | | 13B | | | 30B | | | 65B | | |
|---|---|---|---|---|---|---|---|---|---|---|---|---|---|
| | | PPL (C4) | Lat (s) | Mem (G) | PPL (C4) | Lat (s) | Mem (G) | PPL (C4) | Lat (s) | Mem (G) | PPL (C4) | Lat (s) | Mem (G) |
| Baseline | 16 | 7.08 | 3.2 | 12.7 | 6.61 | 5.6 | 24.6 | 5.98 | OOM | OOM | 5.62 | OOM | OOM |
| GPTQ | 3 | 9.55 | 1.4 | 2.9 | 8.22 | 2.1 | 5.3 | 7.31 | 4.0 | 12.3 | 6.70 | 6.7 | 24.0 |
| SqueezeLLM | 3.02 | 7.75 | 1.5 | 2.9 | 7.08 | 2.4 | 5.4 | 6.37 | 4.0 | 12.5 | 5.99 | 7.6 | 24.5 |
| GPTQ (g128) | 3.25 | 7.89 | 13.7 | 3.0 | 7.12 | 24.2 | 5.6 | 6.47 | 61.9 | 12.9 | 6.01 | 117.8 | 25.1 |
| SqueezeLLM (0.45%) | 3.24 | 7.56 | 1.7 | 3.1 | 6.92 | 2.5 | 5.8 | 6.23 | 4.4 | 14.7 | 5.84 | 8.8 | 28.0 |

While grouping with permutation is an effective way to confine the quantization range, our Dense-and-Sparse scheme can achieve higher accuracy with simpler kernels. We show the latency and peak GPU memory usage of SqueezeLLM in Tab. 3 on an A6000 GPU for different configurations when generating 128 tokens. We observe that the LUT-based non-uniform approach in SqueezeLLM (3rd row) shows up to 2.4× speedup compared to the FP16 baseline, and exhibits comparable latency and peak memory usage to the uniform quantization of non-grouped GPTQ (2nd row). This indicates that the overhead associated with LUT-based dequantization is small, especially considering the considerable perplexity gains it enables.

Additionally, when incorporating sparsity, we still observed latency gains relative to the FP16 baseline. As shown in Tab. 3, keeping 0.45% of parameters in FP16 (4th row) only adds around 10% latency overhead relative to the dense-only implementation, while still resulting in up to 2.2× speed up compared to the FP16 baseline. In contrast, when accounting for permutation, the GPTQ runtime is degraded heavily (5th row). This latency penalty is due to permutation, which means that elements in the same channel need to be scaled using different scaling factors (which are accessed using group indices); it is challenging for these distributed memory accesses to be performed efficiently, as GPUs rely heavily on coalesced memory accesses in order to optimally utilize memory bandwidth. This shows how our Dense-and-Sparse quantization methodology allows for both higher accuracy as well as better performance relative to GPTQ. Additional evaluation results on generating 1024 tokens are provided in Tab. A.3, where we can observe a similar trend.

## 6 CONCLUSION

We have presented SqueezeLLM which attempts to address the Memory Wall problem associated with generative LLM inference that is memory-bound. SqueezeLLM incorporates two novel ideas that allow ultra-low precision quantization of LLMs with negligible degradation in generation performance: the sensitivity-based non-uniform quantization method; and the Dense-and-Sparse decomposition that resolves the outlier issue. We have evaluated SqueezeLLM on a wide range of models and datasets that assess language modeling, problem-solving, and instruction-following capabilities of quantized models, where we have demonstrated that our quantization method can consistently outperform the previous state-of-the-art methodologies.

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
