# A Appendix

## A.1 Related Works on Quantization of Transformer-based Models

Quantization methods can be broadly categorized based whether retraining is required or not Gholami et al. (2021a). Quantization-Aware Training (QAT) requires retraining the model to adapt its weights to help recover accuracy after quantization Zafrir et al. (2019); Shen et al. (2020); Kim et al. (2021); Zhang et al. (2023; 2020); Bai et al. (2020), whereas Post-Training Quantization (PTQ) does not involve retraining Zhao et al. (2019); Cai et al. (2020); Shomron et al. (2021); Oh et al. (2022); Li et al. (2023). While QAT often results in better accuracy, it is often infeasible for LLMs due to the expensive retraining cost and/or lack of access to the training data and infrastructure. As such, most works on LLM quantization have focused on PTQ Yao et al. (2022); Dettmers et al.; Frantar et al. (2022); Yuan et al. (2023); Lin et al. (2023). Our work also focuses on the PTQ approach.

Quantization methods can be also classified as uniform or non-uniform Gholami et al. (2021a). Uniform quantization Frantar et al. (2022); Lin et al. (2023); Dettmers et al. (2023); Zafrir et al. (2019); Shen et al. (2020); Kim et al. (2021); Huang et al. (2023); Liu et al. (2023), which uniformly divides weight ranges into bins, has gained popularity since it allows faster computation by using quantized precision arithmetic. However, recent hardware trends indicate that faster computation does not necessarily translate to improved end-to-end latency or throughput Gholami et al. (2021b), particularly in memory-bound tasks like generative LLM inference (Sec. 3). Furthermore, uniform quantization can be sub-optimal when the weight distribution is non-uniform, as in LLMs (Fig. 2).

Hence, we focus on non-uniform quantization, which non-uniformly allocates quantization bins without constraints for a more accurate representation of weights and smaller quantization errors. While it does not support integer arithmetic for computational acceleration, this drawback is not significant for memory-bound problems as in our focus, where the primary bottleneck lies in memory bandwidth rather than computation. Among non-uniform quantization methods Jeon et al. (2022); Chung et al. (2020), the most similar work to ours is GOBO Zadeh et al. (2020), which introduces a k-means clustering-based look-up table approach. Our work introduces two novel methods as compared to GOBO: (i) sensitivity-aware and (ii) Dense-and-Sparse quantization methodologies, which yield substantial improvements within the k-means-based non-uniform quantization framework.

## A.2 LLaMA Runtime for Different Weight Bit Precision

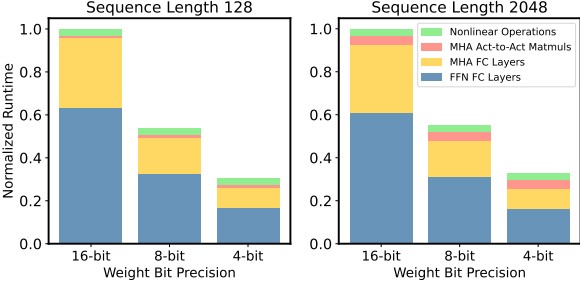

**Figure A.1:** *Normalized runtime for LLaMA-7B when reducing the bit precision for the weights with sequence lengths of 128 (left) and 2048 (right). Results were obtained using a roofline-based performance model for an A5000 GPU. Reducing only the precision of the weights (and not the activations) is sufficient to obtain significant latency reductions.*

## A.3 Experiment Setup (Details)

**Models and Datasets.** We have conducted comprehensive evaluations of SqueezeLLM using various models on different tasks. First, in the language modeling evaluation, we apply SqueezeLLM to the LLaMA Touvron et al. (2023a), LLaMA2 Touvron et al. (2023b) and OPT Zhang et al. (2022) models and measure the perplexity of the quantized models on the C4 Raffel et al. (2020) and Wiki-Text2 Merity et al. (2016) datasets with a chunk size of 2048. We also evaluate the domain-specific knowledge and problem-solving ability through zero-shot MMLU Hendrycks et al. (2021) using the instruction-tuned Vicuna (v1.1 and v1.3) models. We used the Language Model Evaluation Harness

to run zero-shot evaluation across all tasks Gao et al. (2021). Finally, we evaluate the instruction following ability following the methodology presented in Chiang et al. (2023). To do so, we generate answers for 80 sample questions and compared them to the answers generated by the FP16 counterpart using the GPT-4 score. To minimize the ordering effect, we provide the answers to GPT-4 in both orders, resulting in a total of 160 queries.

**Baseline Methods.** We compare SqueezeLLM against PTQ methods for LLMs including RTN as well as state-of-the-art methods including GPTQ Frantar et al. (2022), AWQ Lin et al. (2023) and SpQR Dettmers et al. (2023). To ensure a fair comparison, we use GPTQ *with* activation ordering throughout all experiments unless specified, which addresses the significant performance drop that would otherwise occur. For AWQ, we use official quantized models or reproduce using their official code if they are not available except for LLaMA 65B with group size 256 which ran into OOM even on A100-80G. Evaluations are then conducted based on our perplexity method. For SpQR, we rely on the paper's reported numbers since their perplexity evaluation methodology is identical to ours. SpQR aims to enhance 3-bit and 4-bit models by introducing grouping, bi-level quantization, and sparsity, making them approximately 4 and 4.6 bits on average for LLaMA. In contrast, SqueezeLLM aims to preserve 3 and 4-bit as closely as possible, minimizing any extra model size overhead. Therefore, we present our best-effort comparison of SpQR and SqueezeLLM by comparing 3-bit SpQR models, which average around 4 bits, and our 4-bit models, both of which possess similar model sizes.

**Latency Profiling.** We measure the latency and peak memory usage for generating 128 and 1024 tokens on an A6000 machine using the Torch CUDA profiler. As an official implementation of GPTQ (in particular, the grouped version) is not available, we implement an optimized kernel for single-batch inference based on the most active open-source codebase ( GPTQ-For-LLaMA).

To compare latency with SpQR, we rely on their reported speedup numbers to make our best-effort comparison as their kernel implementation is not publicly available. Regarding AWQ, we utilize the GPTQ kernel without activation ordering since they exhibit identical behavior during inference. Although AWQ has released their own kernel implementation, their 3-bit kernels are not publicly available. Furthermore, they have incorporated optimizations that are unrelated to quantization, such as LayerNorm and positional embedding, which are universally applicable to all quantization methods. To ensure a fair comparison with other methodologies, we refrained from using their released kernels.

### A.4 DATA SKEW IN PER-CHANNEL SPARSITY PATTERN

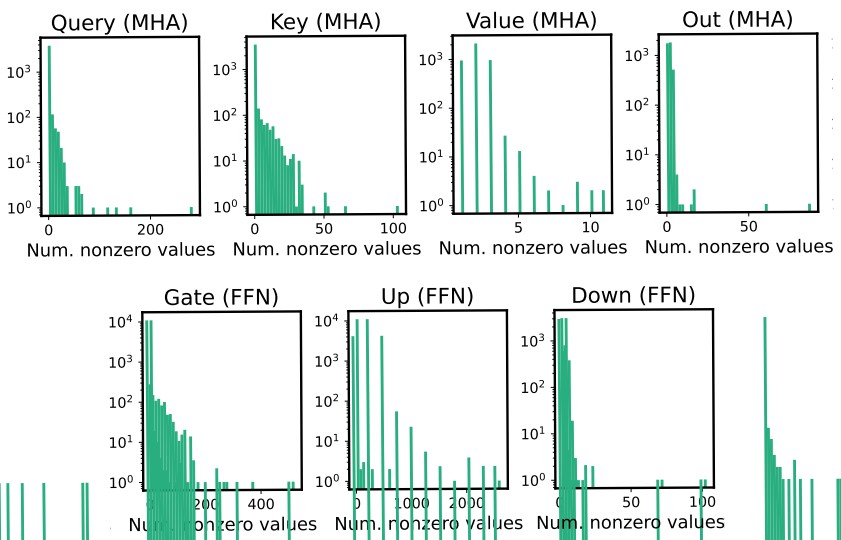

**Figure A.2:** *Histograms of the number of non-zero entries per output channel in 7 different linear layers in the first LLaMA-7B block. The histograms reveal the presence of a few channels that contain significantly more non-zero entries than others, highlighting the skew in the sparsity patterns across different channels within the linear layers.*

**Table A.1:** *Hardware profiling of latency and memory usage for LLaMA 7B, 13B, 30B, and 65B quantized into 3-bit when generating 128 tokens on an A6000 GPU. The first row shows the performance of SqueezeLLM without sparsity. The second row shows the performance of SqueezeLLM with a sparsity level of 0.45% using a standard kernel for processing a CSR matrix. The third row shows the performance of SqueezeLLM with a sparsity level of 0.45% using a balanced sparse kernel that allocates 10 nonzeros per thread, thereby more efficiently handling skewed sparse matrices.*

| Sparse Kernel | Method | Latency (Seconds) | | | | Peak Memory (GB) | | | |
|---|---|---|---|---|---|---|---|---|---|
| | | 7B | 13B | 30B | 65B | 7B | 13B | 30B | 65B |
| | SqueezeLLM (0%) | 1.5 | 2.4 | 4.0 | 7.6 | 2.9 | 5.4 | 12.5 | 24.5 |
| **Standard** | SqueezeLLM (0.45%) | 3.9 | 6.2 | 12.5 | 14.4 | 3.2 | 5.8 | 13.7 | 28.0 |
| **Balanced** | SqueezeLLM (0.45%) | 1.7 | 2.6 | 4.4 | 8.8 | 3.1 | 5.8 | 14.7 | 28.0 |

Fig. A.2 provides the distribution of nonzero entries per output channel across different linear layers in the first LLaMA-7B block. This plot shows that the nonzero distribution is heavily skewed, with a few channels containing a much larger proportion of nonzero values. This skewed distribution makes it challenging to efficiently perform computations using the sparse matrix, as it is difficult to distribute the nonzero elements evenly across parallel processing units. This motivates our modified kernel for handling channels with a large number of outliers in order to reduce the runtime overhead of the sparse matrices. As outlined in Tab. A.1, we observed over 100% added runtime overhead when employing a standard CSR-based kernel. However, if we allocate each thread to process a fixed number of nonzeros (rather than having each thread process an entire row) we were able to drastically reduce the runtime overhead to 10-20% with both sensitive values and outliers.

## A.5 ABLATION STUDIES

### A.5.1 SENSITIVITY-BASED QUANTIZATION.

**Table A.2:** *Ablation study comparing sensitivity-agnostic and sensitivity-based non-uniform quantization on the LLaMA-7B model with 3-bit quantization, measured by perplexity on the C4 benchmark. The baseline model in FP16 achieves a perplexity of 7.08.*

| Method | Sensitivity-Agnostic ($\downarrow$) | Sensitivity-Based ($\downarrow$) |
|---|---|---|
| SqueezeLLM | 18.08 | **7.75** |
| SqueezeLLM (0.05%) | 8.10 | **7.67** |
| SqueezeLLM (0.45%) | 7.61 | **7.56** |

In our ablation study, we investigate the impact of sensitivity-aware weighted clustering on the performance of non-uniform quantization. In Tab. A.2, we compared the performance of sensitivity-aware and sensitivity-agnostic approaches in the context of 3-bit quantization of the LLaMA-7B model. For sensitivity-agnostic quantization, we apply non-weighted k-means clustering at sparsity levels of 0%, 0.05%, and 0.45%. The results demonstrate that while non-uniform quantization alone can reduce the perplexity from 28.26 (of RTN uniform quantization) to 18.08 without considering sensitivity, incorporating sensitivity-aware clustering is critical in reducing the perplexity to 7.75. This improvement is consistent across all sparsity levels.

### A.5.2 IMPACT OF SPARSITY LEVELS ON SQUEEZELLM

In Fig. A.3 (Left), we present the perplexity results of the 3-bit quantized LLaMA-7B model on the C4 benchmarks, with varying percentages of sensitive values extracted as the sparse matrix, ranging from 0% to 0.2%. The plot demonstrates that the perplexity gain diminishes as the sparsity level of the sensitive values exceeds 0.05%. Therefore, we maintain a fixed sparsity level of 0.05% for the sensitive values throughout all experiments.

Furthermore, in Figure A.3 (Right), we compare the performance when the sensitive values are not removed as the sparse matrix (only outlier values are removed) to the case where 0.05% of the sensitive values are removed. In both scenarios, we control the sparsity level by increasing the percentage of outlier values included in the sparse matrix to obtain the trade-off curves. The results

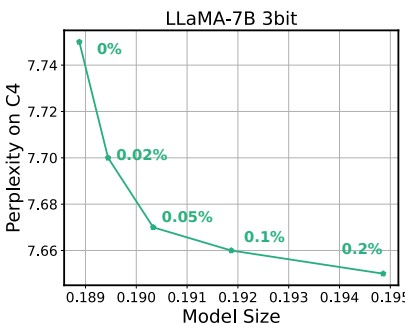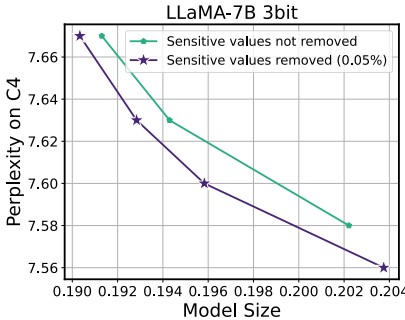

**Figure A.3:** *(Left) Model size (normalized by the size of the FP16 model) and perplexity trade-off with different percentages of sensitive values included in the sparse matrix. Here, no outlier values are included in the sparse matrix. (Right) Comparison of the performance when the sensitive values are not removed as the sparse matrix (only outlier values are removed) to the case where 0.05% of the sensitive values are removed. In both cases, the trade-offs are obtained by controlling the percentage of outlier values included in the sparse matrix.*

indicate that the sparsity configuration with both sensitive values and outlier values consistently outperforms the configuration with only outlier values.

### A.5.3 IMPACT OF GROUPING ON SQUEEZELLM

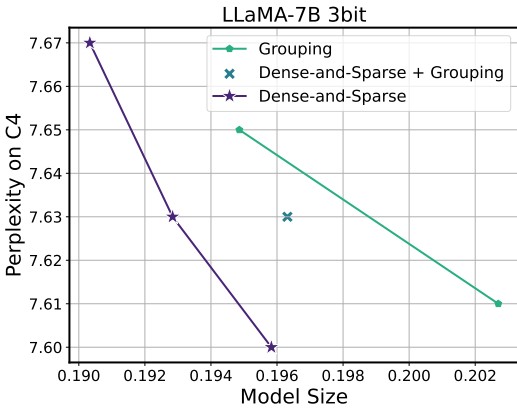

**Figure A.4:** *Model size (normalized by the size of the FP16 model) and perplexity trade-offs of grouping and the Dense-and-Sparse decomposition on 3-bit quantization of the LLaMA-7B model. Here, we compare SqueezeLLM with (i) grouping using group sizes of 1024 and 512 (green), (ii) a hybrid approach that combines a group size of 1024 with a sparsity level of 0.05% (blue), and (iii) the Dense-and-Sparse decomposition approach with varying sparsity levels (violet). The pure Dense-and-Sparse decomposition achieves better size-perplexity trade-offs than both grouping and the hybrid approach.*

In Fig. A.5, we explore the effectiveness of incorporating grouping into SqueezeLLM as an alternative approach to improve quantization performance. We compare three configurations: SqueezeLLM with (i) grouping using group sizes of 1024 and 512 (green), (ii) a hybrid approach that combines a group size of 1024 with a sparsity level of 0.05% (blue), and (iii) the Dense-and-Sparse decomposition approach with varying sparsity levels (violet), where 0.05% of sensitive values are kept and the percentage of outlier values is adjusted. The results clearly demonstrate that both grouping and the hybrid approach result in suboptimal trade-offs compared to the pure Dense-and-Sparse decomposition approach.

This can be attributed to two factors. First, the Dense-and-Sparse decomposition is a direct solution to the outlier issue. In contrast, while grouping can mitigate the impact of outliers to some extent by isolating them within individual groups, it does not provide a direct solution to this issue. In addition,

grouping can introduce significant overhead in terms of storage requirements when combined with non-uniform quantization, since it needs to store one LUT per group. This can be a considerable overhead compared to the uniform quantization approach where only a scaling and zero point value per group need to be stored.

### A.5.4 COMPARISON OF THE OBD FRAMEWORK VERSUS THE OBS FRAMEWORK FOR NON-UNIFORM QUANTIZATION

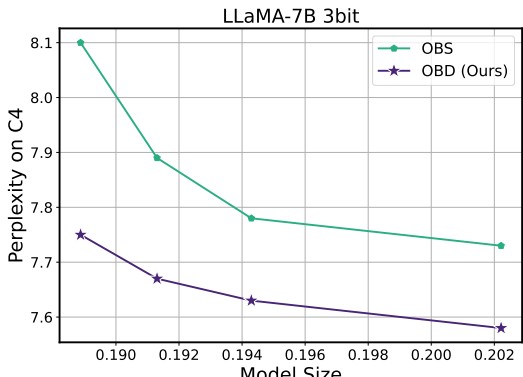

**Figure A.5:** *Model size (normalized by the size of the FP16 model) and perplexity trade-offs for 3-bit quantization of the LLaMA-7B model using the Optimal Brain Surgeon (OBS) framework versus the Optimal Brain Damage (OBD) framework for determining the non-uniform quantization configuration. The trade-off is obtained by adjusting the sparsity level of the outliers being extracted. Across all sparsity levels, the OBD framework, which is the foundation for SqueezeLLM, consistently outperforms the OBS framework as an alternative approach.*

While our method adopts the Optimal Brain Damage (OBD) framework to minimize the perturbation of the final output of the model during quantization, it is worth noting that the Optimal Brain Surgeon (OBS) framework can also be considered as an alternative. Most existing solutions for LLM quantization including GPTQ Frantar et al. (2022), AWQ Lin et al. (2023), and SpQR Dettmers et al. (2023) have utilized the OBS framework, which aims to minimize the perturbation of output activations in individual layers. In this ablation study, we demonstrate that the OBD framework is superior to the OBS framework.

Under the OBD framework, the optimization objective for determining the non-uniform quantization configuration can be reformulated as $\arg\min_Q \|WX - W_Q X\|_2^2$, where $X$ denotes a batch of input activations. This object can be approximated as a weighted k-means clustering problem, where each weight is weighted by the square of the corresponding input activation size. This indeed results in the activation-based sensitivity/importance metric as in the AWQ framework Lin et al. (2023).

In Fig. A.5.4, we compare the perplexity on the C4 dataset for 3-bit quantization of the LLaMA-7B model using the OBS framework versus the OBD framework. Across all sparsity levels obtained by adjusting the number of outliers being extracted, SqueezeLLM based on the OBD framework outperforms the alternative of using the OBS framework by a large margin of up to around 0.3 perplexity points.

### A.6 ADDITIONAL HARDWARE PROFILING RESULTS

In Tab. A.3, we provide additional hardware profiling results using a sequence length of 1024. All the experimental setups and details are identical to Sec. 5.4 and Tab. 3.

**Table A.3:** *Latency (s) and peak memory usage (GB) of 3-bit LLaMA when generating 1024 tokens on an A6000 GPU. The table compares the FP16 baseline, non-grouped and grouped GPTQ with activation ordering, and SqueezeLLM with different sparsity levels. For comparison, we include bitwidth and perplexity on the C4 benchmark.*

| Method | Bit width | 7B | | | 13B | | | 30B | | | 65B | | |
|---|---|---|---|---|---|---|---|---|---|---|---|---|---|
| | | PPL (C4) | Lat (s) | Mem (G) | PPL (C4) | Lat (s) | Mem (G) | PPL (C4) | Lat (s) | Mem (G) | PPL (C4) | Lat (s) | Mem (G) |
| Baseline | 16 | 7.08 | 26.5 | 13.1 | 6.61 | 47.0 | 25.2 | 5.98 | OOM | OOM | 5.62 | OOM | OOM |
| GPTQ | 3 | 7.55 | 12.6 | 3.3 | 6.22 | 19.1 | 6.0 | 5.76 | 36.8 | 13.8 | 5.58 | 60.2 | 26.2 |
| SqueezeLLM | 3.02 | 6.32 | 13.6 | 3.4 | 5.60 | 21.2 | 6.1 | 4.66 | 37.8 | 16.1 | 4.05 | 66.9 | 29.9 |
| GPTQ (g128) | 3.25 | 6.27 | 110.7 | 3.4 | 5.47 | 176.1 | 6.2 | 4.83 | 500.8 | 14.3 | 4.55 | 955.2 | 27.3 |
| SqueezeLLM (0.45%) | 3.24 | 6.13 | 14.6 | 3.6 | 5.45 | 22.2 | 6.5 | 4.44 | 42.5 | 17.4 | 3.88 | 82.35 | 32.4 |

**Table A.4:** *Perplexity comparison of LLaMA-30B and 65B models quantized into 4 and 3 bits using different methods including RTN, GPTQ, AWQ and SpQR on C4 and WikiText-2. We compare the performance of GPTQ, AWQ, and SqueezeLLM (SQLLM) in groups based on similar model sizes. In the first group, we compare dense-only SqueezeLLM with non-grouped GPTQ. In the subsequent groups, we compare SqueezeLLM with different levels of sparsity to GPTQ and AWQ with different group sizes.*

| LLaMA-30B | 3-bit | | | 4-bit | | | LLaMA-65B | 3-bit | | | 4-bit | | |
|---|---|---|---|---|---|---|---|---|---|---|---|---|---|
| Method | Avg. Bits (comp. rate) | PPL (↓) C4 | Wiki | Avg. Bits (comp. rate) | PPL (↓) C4 | Wiki | Method | Avg. Bits (comp. rate) | PPL (↓) C4 | Wiki | Avg. Bits (comp. rate) | PPL (↓) C4 | Wiki |
| Baseline | 16 | 5.98 | 4.10 | 16 | 5.98 | 4.10 | Baseline | 16 | 5.62 | 3.53 | 16 | 5.62 | 3.53 |
| RTN | 3 (5.33) | 28.53 | 14.89 | 4 (4.00) | 6.33 | 4.54 | RTN | 3 (5.33) | 12.77 | 10.59 | 4 (4.00) | 5.86 | 3.92 |
| GPTQ | 3 (5.33) | 7.31 | 5.76 | 4 (4.00) | 6.20 | 4.43 | GPTQ | 3 (5.33) | 6.70 | 5.58 | 4 (4.00) | 5.81 | 4.11 |
| SpQR | - | - | - | 3.89 (4.11) | 6.08 | 4.25 | SpQR | 3 (5.33) | - | 4.2[†] | 3.90 (4.10) | 5.70 | **3.68** |
| SQLLM | 3.02 (5.31) | **6.37** | **4.66** | 4.03 (3.97) | **6.06** | **4.22** | SQLLM | 3.02 (5.30) | **5.99** | **4.05** | 4.04 (3.96) | **5.69** | 3.76 |
| GPTQ (g128) | 3.25 (4.92) | 6.47 | 4.83 | 4.25 (3.77) | 6.07 | 4.24 | GPTQ (g128) | 3.25 (4.92) | 6.01 | 4.55 | 4.25 (3.77) | 5.69 | 3.76 |
| AWQ (g128) | 3.25 (4.92) | 6.38 | 4.63 | 4.25 (3.77) | 6.05 | 4.21 | AWQ (g128) | 3.25 (4.92) | 5.94 | 4.00 | 4.25 (3.77) | 5.68 | 3.67 |
| SQLLM (0.45%) | 3.25 (4.92) | **6.23** | **4.44** | 4.25 (3.77) | **6.04** | **4.18** | SQLLM (0.45%) | 3.24 (4.94) | **5.84** | **3.88** | 4.26 (3.76) | **5.67** | **3.63** |

**Table A.5:** *Perplexity comparison of LLaMA2 models quantized into 4 and 3 bits using different methods including RTN, GPTQ, AWQ and SpQR on C4 and WikiText-2. We compare the performance of GPTQ, AWQ, and SqueezeLLM (SQLLM) in groups based on similar model sizes. In the first group, we compare dense-only SqueezeLLM with non-grouped GPTQ. In the subsequent groups, we compare SqueezeLLM with different levels of sparsity to GPTQ and AWQ with different group sizes. Note that all GPTQ results are with activation reordering.*

| LLaMA2-7B | 3-bit | | | 4-bit | | | LLaMA2-13B | 3-bit | | | 4-bit | | |
|---|---|---|---|---|---|---|---|---|---|---|---|---|---|
| Method | Avg. Bits (comp. rate) | PPL (↓) C4 | Wiki | Avg. Bits (comp. rate) | PPL (↓) C4 | Wiki | Method | Avg. Bits (comp. rate) | PPL (↓) C4 | Wiki | Avg. Bits (comp. rate) | PPL (↓) C4 | Wiki |
| Baseline | 16 | 6.97 | 5.47 | 16 | 6.97 | 5.47 | Baseline | 16 | 6.47 | 4.88 | 16 | 6.47 | 4.88 |
| RTN | 3 (5.33) | 404.45 | 542.86 | 4 (4.00) | 7.72 | 6.12 | RTN | 3 (5.33) | 12.50 | 10.68 | 4 (4.00) | 6.83 | 5.20 |
| GPTQ | 3 (5.33) | 10.45 | 8.97 | 4 (4.00) | 7.42 | 5.90 | GPTQ | 3 (5.33) | 8.27 | 6.17 | 4 (4.00) | 6.74 | 5.08 |
| SQLLM | 3.02 (5.29) | **7.72** | **6.18** | 4.05 (3.95) | **7.12** | **5.62** | SQLLM | 3.02 (5.30) | **6.97** | **5.36** | 4.04 (3.96) | **6.57** | **4.99** |
| GPTQ (g128) | 3.24 (4.93) | 7.97 | 6.25 | 4.24 (3.77) | 7.23 | 5.72 | GPTQ (g128) | 3.25 (4.92) | 7.06 | 5.31 | 4.25 (3.77) | 6.57 | 4.96 |
| AWQ (g128) | 3.24 (4.93) | 7.84 | 6.24 | 4.24 (3.77) | 7.13 | 5.72 | AWQ (g128) | 3.25 (4.92) | 6.94 | 5.32 | 4.25 (3.77) | 6.56 | 4.97 |
| SQLLM (0.45%) | 3.24 (4.93) | **7.51** | **5.96** | 4.27 (3.75) | **7.08** | **5.57** | SQLLM (0.45%) | 3.24 (4.94) | **6.82** | **5.23** | 4.26 (3.76) | **6.54** | **4.96** |

## A.7 ADDITIONAL EXPERIMENT RESULTS

### A.7.1 PERPLEXITY EVALUATION

In Tab. A.4, we provide the full experimental results on LLaMA Touvron et al. (2023a). Furthermore, in Tab. A.5 and A.6, we provide additional experimental results on LLaMA2 Touvron et al. (2023b) and OPT Zhang et al. (2022) models.

---

[†]SpQR does not report their near-3-bit performance. However, in the case of 65B model, its 3-bit perplexity on Wikitext-2 can be inferred from the trade-off curve in Figure 8 of their paper. This comparison indicates that the gap between SpQR and SqueezeLLM can be larger in the lower-bitwidth regimes.

**Table A.6:** *Perplexity comparison of OPT models quantized into 4 and 3 bits using different methods including RTN, GPTQ, AWQ and SpQR on C4 and WikiText-2. We compare the performance of GPTQ, AWQ, and SqueezeLLM (SQLLM) in groups based on similar model sizes. In the first group, we compare dense-only SqueezeLLM with non-grouped GPTQ. In the subsequent groups, we compare SqueezeLLM with different levels of sparsity to GPTQ and AWQ with different group sizes. Note that all GPTQ results are with activation reordering. "div" means that the perplexity is diverged.*

| OPT-1.3B | 3-bit | | | 4-bit | | |
|---|---|---|---|---|---|---|
| **Method** | **Avg. Bits** (comp. rate) | **PPL** (↓) C4 | Wiki | **Avg. Bits** (comp. rate) | **PPL** (↓) C4 | Wiki |
| Baseline | 16 | 14.72 | 14.62 | 16 | 14.72 | 14.62 |
| RTN | 3 (5.43) | div. | div. | 4 (4) | 24.68 | 48.19 |
| SQLLM | 3.04 (5.26) | **16.42** | **16.30** | 4.09 (3.91) | **15.01** | **14.94** |
| AWQ (g128) | 3.25 (4.93) | 16.28 | 16.32 | 4.25 (3.77) | 15.04 | 14.95 |
| SQLLM (0.5%) | 3.25 (4.92) | **15.84** | **15.76** | 4.30 (3.72) | **14.94** | **14.83** |

| OPT-2.7B | 3-bit | | | 4-bit | | |
|---|---|---|---|---|---|---|
| **Method** | **Avg. Bits** (comp. rate) | **PPL** (↓) C4 | Wiki | **Avg. Bits** (comp. rate) | **PPL** (↓) C4 | Wiki |
| Baseline | 16 | 13.17 | 12.47 | 16 | 13.17 | 12.47 |
| RTN | 3 (5.33) | div. | div. | 4 (4) | 17.52 | 16.92 |
| SQLLM | 3.04 (5.26) | **14.45** | **13.85** | 4.07 (3.93) | **13.38** | **12.80** |
| AWQ (g128) | 3.25 (4.93) | 16.28 | 16.32 | 4.25 (3.77) | 13.39 | 12.73 |
| SQLLM (0.5%) | 3.25 (4.92) | **13.88** | **13.43** | 4.29 (3.73) | **13.30** | **12.60** |

| OPT-6.7B | 3-bit | | | 4-bit | | |
|---|---|---|---|---|---|---|
| **Method** | **Avg. Bits** (comp. rate) | **PPL** (↓) C4 | Wiki | **Avg. Bits** (comp. rate) | **PPL** (↓) C4 | Wiki |
| Baseline | 16 | 11.74 | 10.86 | 16 | 11.74 | 10.86 |
| RTN | 3 (5.33) | div. | div. | 4 (4) | 13.38 | 12.10 |
| SpQR | - | - | - | 3.94 (4.06) | 11.98 | 11.04 |
| SQLLM | 3.02 (5.29) | **12.44** | **11.70** | 4.05 (3.96) | **11.85** | **11.03** |
| SpQR | - | - | - | 4.27 (3.74) | 11.88 | **10.91** |
| AWQ (g128) | 3.25 (4.92) | 12.30 | 11.41 | 4.25 (3.77) | 11.86 | 10.93 |
| SQLLM (0.5%) | 3.26 (4.90) | **12.18** | **11.31** | 4.28 (3.73) | **11.83** | 10.92 |

| OPT-13B | 3-bit | | | 4-bit | | |
|---|---|---|---|---|---|---|
| **Method** | **Avg. Bits** (comp. rate) | **PPL** (↓) C4 | Wiki | **Avg. Bits** (comp. rate) | **PPL** (↓) C4 | Wiki |
| Baseline | 16 | 11.20 | 10.12 | 16 | 11.20 | 10.12 |
| RTN | 3 (5.33) | div. | div. | 4 (4) | 12.35 | 11.32 |
| SpQR | - | - | - | 3.93 (4.07) | 11.34 | 10.28 |
| SQLLM | 3.02 (5.29) | **12.69** | **11.76** | 4.05 (3.96) | **11.29** | **10.24** |
| SpQR | - | - | - | 4.27 (3.74) | 11.27 | **10.22** |
| AWQ (g128) | 3.25 (4.92) | 12.61 | 10.67 | 4.25 (3.77) | 11.28 | **10.22** |
| SQLLM (0.5%) | 3.26 (4.90) | **11.57** | **10.54** | 4.28 (3.73) | **11.26** | **10.22** |

| OPT-30B | 3-bit | | | 4-bit | | |
|---|---|---|---|---|---|---|
| **Method** | **Avg. Bits** (comp. rate) | **PPL** (↓) C4 | Wiki | **Avg. Bits** (comp. rate) | **PPL** (↓) C4 | Wiki |
| Baseline | 16 | 10.69 | 9.56 | 16 | 10.69 | 9.56 |
| RTN | 3 (5.33) | div. | div. | 4 (4) | 11.90 | 10.98 |
| SpQR | - | - | - | 3.94 (4.06) | 10.78 | **9.54** |
| SQLLM | 3.01 (5.31) | **11.10** | **10.17** | 4.03 (3.97) | **10.75** | 9.65 |
| SpQR | - | - | - | 4.26 (3.76) | 10.73 | **9.50** |
| AWQ (g128) | 3.25 (4.92) | 10.96 | 9.85 | 4.25 (3.77) | 10.75 | 9.59 |
| SQLLM (0.5%) | 3.26 (4.90) | **10.93** | **9.77** | 4.28 (3.73) | **10.72** | 9.61 |

**Table A.7:** *Comparison of PTQ methods on zero-shot MMLU accuracy applied to Vicuna v1.3.*

| Method | Avg. Bit | 7B (↑) | 13B (↑) | 33B (↑) |
|---|---|---|---|---|
| Baseline | 16 | 40.2% | 43.3% | 50.4% |
| AWQ (g128) | 4.25 | **39.6%** | 42.2% | 49.5% |
| SqueezeLLM | 4.05 | 39.3% | **44.1%** | 48.0% |
| SqueezeLLM (0.45%) | 4.26 | 39.5% | 43.8% | **49.9%** |
| AWQ (g128) | 3.25 | 37.4% | 40.7% | 46.4% |
| SqueezeLLM | 3.02 | 35.1% | 40.5% | 46.2% |
| SqueezeLLM (0.45%) | 3.24 | **37.6%** | **40.8%** | **47.7%** |

### A.7.2 MMLU Evaluation

In Tab. A.7, we provide additional experimental results for Vicuna v1.3 on MMLU.

### A.8 Limitations

While our empirical results primarily focus on generation tasks, the proposed ideas in this work are not inherently limited to decoder architectures. However, we have not yet conducted thorough assessments of our framework's effectiveness on encoder-only or encoder-decoder architectures, as well as other neural network architectures. Additionally, it is important to note that our hardware performance modeling approach relies on a simulation-based method using a roofline model, which entails making simplified assumptions about the hardware's inference pipeline.