# OpenReview forum: "SqueezeLLM: Dense and Sparse Quantization"
_ICLR.cc/2024/Conference — Submitted to ICLR 2024_

### Official Review · Reviewer_7Q1V · 2023-10-27

**Soundness:** 4 excellent
**Presentation:** 3 good
**Contribution:** 2 fair
**Rating:** 8
**Confidence:** 4

**Summary:**

In this article, the authors propose to address the memory footprint of LLMs. They argue that this represents the main challenge for efficient inference of such models. This claim is well-supported by previous literature in the field. To do so, they propose to combine two main elements: sparse encoding of outliers and hessian-based clustering. The presence of outliers is a well-known challenge regarding LLM quantization. These weights are challenging to encode. Instead of occupying values within the non-uniform quantization codebook. The author decompose the weight tensor in two: a small size (less than 1% of the total) is dedicated to said outliers in high precision and the remainder of the weight values are encoded in a low size LUT. Furthermore, in order to improve the performance of the LUT clustering, the authors propose to adapt the k-means algorithm to the specificity of the weight tensors of a trained model: all weight values are not trained equal. The hessian of the weight values is leveraged as an estimate to weight each weights contribution.
The resulting method achieves remarkable results on a variety of challenging benchmarks.

**Strengths:**

The paper is well-written, the proposed method performs well and is thoroughly benchmarked on challenging problems. Although the core elements are not completely novel, from the outliers encoding to the use of the Hessian for importance estimation, the combination of the two is well organized and not so trivial. Consequently, this research contains all the necessary element for a paper published at ICLR.

**Weaknesses:**

Overall, I have two remarks for improvement. First, here are some missing references. Regarding sparse encoding of the outliers, [1] propose an approach that bears some similarities and would be worth mentioning. Similarly, regarding the measurement of the importance using the Hessian matrix, pruning techniques have previously used similar estimates [2,3] and in particular [4]. I think the paper would benefit from these elements. Second, I think the results are great, so there is no need to not highlight other methods when they perform on par with the proposed SqueezeLLM.

[1] Yvinec, Edouard, et al. "REx: Data-Free Residual Quantization Error Expansion." arXiv preprint arXiv:2203.14645 (2022).
[2] Molchanov, Pavlo, et al. "Importance estimation for neural network pruning." Proceedings of the IEEE/CVF conference on computer vision and pattern recognition. 2019.
[3] Li, Mingchen, et al. "Exploring weight importance and hessian bias in model pruning." arXiv preprint arXiv:2006.10903 (2020).
[4] Yu, Shixing, et al. "Hessian-aware pruning and optimal neural implant." Proceedings of the IEEE/CVF Winter Conference on Applications of Computer Vision. 2022.

**Questions:**

I wonder how would the authors fine-tune such compressed model without losing the benefits of the compression technique, i.e. how to fold a LoRA or else.

---

> ### Author Response · Authors · 2023-11-16
>
> ## W1. References
>
>
> > First, here are some missing references. Regarding sparse encoding of the outliers, [1] propose an approach that bears some similarities and would be worth mentioning. Similarly, regarding the measurement of the importance using the Hessian matrix, pruning techniques have previously used similar estimates [2,3] and in particular [4]. I think the paper would benefit from these elements. Second, I think the results are great, so there is no need to not highlight other methods when they perform on par with the proposed SqueezeLLM.
>
> We appreciate the reviewer’s valuable feedback and the key references. We will include these references and comparisons in the final version of our paper to provide a more comprehensive context.
>
> ---
>
> ## Q1. Ideas on Fine-tuning
>
>
> > I wonder how would the authors fine-tune such compressed model without losing the benefits of the compression technique, i.e. how to fold a LoRA or else.
>
> This is indeed an interesting question given that finetuning LLMs is emerging as an interesting research field. We envision that SqueezeLLM could be effectively employed as a backbone quantization method in parameter-efficient finetuning scenarios similar to QLoRA [1] – where the main model would be quantized for the forward pass, and only the smaller adapter modules would undergo weight updates – without losing the benefits of our compression technique. While it's common in practice [1, 2] to retain the LoRA module after fine-tuning, how to fold them will be an interesting avenue for future research.
>
> [1] Dettmers, Tim, et al. "Qlora: Efficient finetuning of quantized llms." *arXiv preprint arXiv:2305.14314* (2023).
>
> [2] Chai, Yuji, et al. "INT2. 1: Towards Fine-Tunable Quantized Large Language Models with Error Correction through Low-Rank Adaptation." *arXiv preprint arXiv:2306.08162* (2023).

---

> > ### Author Response · Authors · 2023-11-22
> >
> > Dear Reviewer 7Q1V
> > \
> > \
> > We hope that our response has provided clarification to your concerns.
> > As today is the last day for author-reviewer communication, we would greatly appreciate it if you could please let us know if we could provide any further clarifications about the paper.
> >
> > Thanks so much again for taking the time to review our paper.
> > \
> > \
> > Best regards,
> >
> > SqueezeLLM Authors

---

> > > ### Comment · Reviewer_7Q1V · 2023-11-23
> > > **response**
> > >
> > > I can confirm that the authors addressed my concerns. Consequently, I'll keep my initial rating.

---

### Official Review · Reviewer_sqPk · 2023-10-31

**Soundness:** 3 good
**Presentation:** 3 good
**Contribution:** 2 fair
**Rating:** 5
**Confidence:** 4

**Summary:**

The paper aims to demonstrate that the main challenge with single-batch inference in generative Large Language Models (LLMs) is memory bandwidth rather than computing. The authors propose SqueezeLLM, a post-training quantization framework that can compress LLMs to 3-bit without losing performance. It uses the following ideas to optimize each weight tensor's bit precision and memory usage.

- **Sensitivity-based non-uniform quantization**: The optimal bit precision assignment for each weight tensor can be determined by its sensitivity to quantization error, approximated by second-order information.
- **Dense-and-Sparse decomposition**:  The outliers and sensitive weight values that cannot be quantized effectively can be stored in a sparse format with full precision. The remaining weight values correspond to dense components that could be easily quantized.

**Strengths:**

The paper is well-written, and the contributions are easy to understand. Figure 1 illustrates the impact of the key approaches of the work. The research community, spanning hardware and ML, has well-acknowledged the memory wall issue for such models. Section 3 provides this line of work justice with supporting illustration in Fig. A.1.

The paper is well positioned in related works – focus on weight quantization versus activation, following the OBD vs OBC (GPTQ Frantar et al. (2022), AWQ Lin et al. (2023), and SpQR Dettmers et al. (2023)) framework to preserve final output performance.

For applications such as LLMs, the authors combine two techniques: K-means clustering for quantization of sensitive weights/outliers guided by second-order derivative and dense and sparse decomposition/quantization. Please refer to the Weakness section for further discussion.

The experiments and empirical analysis are extensive, supporting the technical contributions.

**Weaknesses:**

Regarding the two approaches, there needs to be more discussion on prior works.
K-means clustering for Quantization is a popular technique used in signal processing. For neural networks, works such as DeepCompression [Han et al., ICLR 2016], SLQ/MLQ [Xu et al., AAAI 2018], HPTQ [Xu et al., IEEE Signal Processing Letters 2023] employ k-means clustering for quantization. It would be great to acknowledge such and similar works in literature and compare them to justify the novelty claimed in Sec 4.1. Of course, LLMs as a target application is new, but the technique may need to be more novel in its current presentation.

A similar treatment is observed in Dense and Spare Decomposition (Sec 4.2), where similar techniques exist in the literature, not necessarily for LLMs.  The author must acknowledge and discuss the prior works, such as DSD [Han et al., ICLR 2017], Scatterbrain [Chen et al., NeurIPS 2021], Vitality [Dass et al., HPCA 2023], etc to bring out the novelty.

Overall, it seems the LLMs provide opportunities to employ a novel combination of existing techniques. If so, the current presentation of Sec 4 does not paint an accurate picture and needs to be rewritten to acknowledge similar works and present the novelty. In addition, discussing SmoothQuant [Xiao et al., arXiv 2022, ICML 2023] in the context of post-training quantization for LLMs might be worthwhile.

**Questions:**

As discussed in the Weakness section, the authors should clarify the novelty by acknowledging and discussing prior works with the techniques presented. Are the techniques novel, or do LLMs provide opportunities for a novel combination of the existing techniques in literature? Sec 4, in its presentation, implies that techniques are novel, which may not be necessarily true.

---

> ### Author Response · Authors · 2023-11-16
> **Official Comment by Authors [1/2]**
>
> We appreciate the reviewer's insightful comments and the references provided. We have carefully reviewed the mentioned works and emphasize our novelty as follows. We will include this comparison in the paper to acknowledge the prior works and clarify the novelty of our approach.
>
> ---
>
> ## W1. Comparisons with Other Clustering Methods
>
>
> > Regarding the two approaches, there needs to be more discussion on prior works. K-means clustering for Quantization is a popular technique used in signal processing. For neural networks, works such as DeepCompression [Han et al., ICLR 2016], SLQ/MLQ [Xu et al., AAAI 2018], HPTQ [Xu et al., IEEE Signal Processing Letters 2023] employ k-means clustering for quantization. It would be great to acknowledge such and similar works in literature and compare them to justify the novelty claimed in Sec 4.1. Of course, LLMs as a target application is new, but the technique may need to be more novel in its current presentation.
>
> That is a fair question. About the comparison with prior k-means methods, we kindly emphasize that our approach is different. In particular, SqueezeLLM incorporates a **sensitivity-aware weighted k-means clustering method combined with the dense-and-sparse decomposition, which allows for effective low-bit LLM quantization.**
>
> In particular, our *sensitivity-aware* weighted clustering is different from the non-weighted (i.e. sensitivity-agnostic) clustering approaches in the prior works and allows minimal performance degradation without retraining. As discussed in Section 4.1, our method ensures minimal loss degradation under widely accepted assumptions, and therefore significantly reduces performance degradation in contrast to the sensitivity-agnostic approach. This is demonstrated in Figure 1, with a considerable improvement from 18.09 (sensitivity-agnostic) to 7.75 (sensitivity-aware). Furthermore, such a precise allocation of LUT entries through the sensitivity-aware approach is especially critical for LLM quantization, where retraining or finetuning can be highly restricted. This is in contrast with methodologies like DeepCompression or SLQ/MLQ, which performs post-clustering retraining to recover performance.
>
> ---
>
> ## W2. Comparisons with Other Decomposition Methods
>
>
> > A similar treatment is observed in Dense and Sparse Decomposition (Sec 4.2), where similar techniques exist in the literature, not necessarily for LLMs. The author must acknowledge and discuss the prior works, such as DSD [Han et al., ICLR 2017], Scatterbrain [Chen et al., NeurIPS 2021], Vitality [Dass et al., HPCA 2023], etc to bring out the novelty.
>
> DSD is a significant paper that introduces a novel way of training neural networks where the term "sparse" represents *post-pruned weights*, and "dense" denotes weights that have been recovered post-pruning. However, in our approach, “sparse” refers to isolated *outlier weights* to enhance post-training quantization performance. Furthermore, DSD operates as a *training-time technique*, which trains the post-pruned (so-called sparse) weights and the post-recovery (so-called dense) components separately before merging them into a single matrix, whereas our method is tailored for *inference*, with a major focus on enhancing inference-time efficiency and latency. While DSD and ours use the same terms of “dense” and “sparse”, their implications are dissimilar.
>
> With regards to Scatterbrain and Vitality, our method proposes a direct solution to tackle the outlier issue in LLMs, which has been a major obstacle for low-bit LLM quantization. In contrast, both Scatterbrain and Vitality introduce sparsity to represent *attention maps* as a combination of large attention values and the low-rank approximation of the residual matrix. Additionally, our approach further incorporates “sensitive” values within the sparse matrix, which significantly mitigates distortion in non-uniform quantization LUT entries, leading to a considerable improvement in post-quantization performance (7.67 vs 7.56 in Figure 1).

---

> ### Author Response · Authors · 2023-11-16
> **Official Comment by Authors [2/2]**
>
> ## W3. Novelty and Comparison with SmoothQuant
>
>
> > Overall, it seems the LLMs provide opportunities to employ a novel combination of existing techniques. If so, the current presentation of Sec 4 does not paint an accurate picture and needs to be rewritten to acknowledge similar works and present the novelty. In addition, discussing SmoothQuant [Xiao et al., arXiv 2022, ICML 2023] in the context of post-training quantization for LLMs might be worthwhile.
>
> LLMs indeed present *unique behavior and challenges* compared to previous models, necessitating novel approaches to enable effective low-bit quantization. SmoothQuant, as the reviewer mentioned, exemplifies one such novel approach that offers a quantization method that tackles new challenges in LLMs and facilitates more accurate quantization. We would also like to kindly emphasize that SqueezeLLM has introduced two novel methods, the sensitivity-aware weighted k-means clustering method combined with the dense-and-sparse decomposition, to tackle the *unique* challenges in effective low-bit LLM quantization.
>
> We acknowledge the reviewer’s suggestion, and we will include SmoothQuant in the paper. At the same time, we kindly emphasize that our approach is fundamentally different from SmoothQuant. While SmoothQuant is a quantization framework for both *weights and activations* with a specific focus on mitigating the *activation outliers* by reweighting the associated weight channels, SqueezeLLM is a *weight-only* quantization method that focuses on *weight outliers*.

---

> > ### Author Response · Authors · 2023-11-22
> >
> > Dear Reviewer sqPk
> > \
> > \
> > We hope that our response has provided clarification to your concerns.
> > As today is the last day for author-reviewer communication, we would greatly appreciate it if you could please let us know if we could provide any further clarifications about the paper.
> >
> > Thanks so much again for taking the time to review our paper.
> > \
> > \
> > Best regards,
> >
> > SqueezeLLM Authors

---

### Official Review · Reviewer_Ed31 · 2023-11-01

**Soundness:** 2 fair
**Presentation:** 2 fair
**Contribution:** 2 fair
**Rating:** 5
**Confidence:** 4

**Summary:**

This study introduces a novel method for compressing memory-limited LLMs by employing two primary techniques: 1) non-uniform quantization, which is based on clustering, and 2) the identification and extraction of outliers within weight parameters. While these strategies have the potential to disrupt acceleration mechanisms in high-performance computing systems, this paper also offers a dedicated kernel tailored for the proposed method. The efficacy of both the method and the kernel is further demonstrated through comprehensive experimental results on contemporary LLMs.

**Strengths:**

- This paper is articulately composed and effectively describes the memory challenges associated with LLMs.
- The study adeptly builds upon existing compression techniques for LLMs. While many papers focus solely on uniform quantization, this one introduces non-uniform quantization.
- The experimental results encompass a wide range of models and datasets.

**Weaknesses:**

*Concerns Regarding Citations:*
The citation format utilized in the paper is incorrect. I'd recommend adhering to the ICLR latex format for consistency.
I noticed references to non-uniform quantization, specifically (Chung, 2020) and (Jeon, 2022), in the appendix. These papers employ non-uniform quantization as a structured extension of binary quantization. This differs from the clustering-based vector quantization used in your paper.

*Implementation and Kernel Concerns:*
I have reservations about the kernel implementation. Introducing such custom kernel, as described in your method, appears to disrupt the established high-performance computing framework. Although an unstructured or non-uniform structure might enhance model performance, it could also negatively impact acceleration performance or complicate it. Hence, a more detailed exposition about the kernel is essential to substantiate the novelty of your method. While your results indicate an improved model performance, the absence of a dedicated kernel might undermine its effectiveness. Additionally, it's worth pondering why there was no effort to implement and optimize the kernel specifically for the A100. Even though the A100/H100 might be a costly choice, they can serve as a foundational hardware for large language models, particularly in tandem with NVIDIA's inference software. It's noteworthy that the memory bandwidth for the RTX series is below 1TB/s.

*Feedback on Model Performance:*
The majority of the model performance results focus on PPL outcomes. I believe it would be beneficial to include MMLU or CSR results for larger models within the main content, rather than relegating it to the appendix.

In conclusion, due to the aforementioned concerns, I am inclined to assign a score of 5 to this paper. While I perceive this paper as integrating various methodologies, I will defer to other reviewers for their viewpoints on this aspect.

**Questions:**

included in weaknesses

---

> ### Author Response · Authors · 2023-11-16
>
> We appreciate the reviewer's insightful comments, and here we address your comments in detail:
>
> ## W1. Kernel Concerns and Evaluation on A100
>
> > Implementation and Kernel Concerns: I have reservations about the kernel implementation. Introducing such a custom kernel, as described in your method, appears to disrupt the established high-performance computing framework. Although an unstructured or non-uniform structure might enhance model performance, it could also negatively impact acceleration performance or complicate it. Hence, a more detailed exposition about the kernel is essential to substantiate the novelty of your method. While your results indicate an improved model performance, the absence of a dedicated kernel might undermine its effectiveness. Additionally, it's worth pondering why there was no effort to implement and optimize the kernel specifically for the A100. Even though the A100/H100 might be a costly choice, they can serve as a foundational hardware for large language models, particularly in tandem with NVIDIA's inference software. It's noteworthy that the memory bandwidth for the RTX series is below 1TB/s.
>
> This is a fair point. We had originally focused on lower-grade GPUs partly due to the need for enabling inference on more affordable GPUs, and partially due to our limited access to A100 systems for our experiments. However, in light of the reviewer’s insightful comments, we have conducted additional experiments using our kernels on an A100 system to run the linear layers of LLaMA model for generating a sequence length of 128. As demonstrated in the table below, our kernel implementation still attains **1.5-2.5x performance speedups** relative to the fp16 matrix-vector multiply kernel across different model sizes, similar to what we have observed with our previous experiments. This is despite any additional optimizations or tuning.
>
> Therefore, our evaluation results indicate that SqueezeLLM **does not introduce any negative impact on acceleration performance** due to our unstructured or non-uniform structure. Furthermore, we would like to highlight that the implementation of SqueezeLLM does not introduce complexity. The core of SqueezeLLM is non-uniform quantization which only involves look-up-table operations, which can be implemented as *simply and efficiently* as those in uniform quantization.
> \
> \
> **Matrix-Vector Multiply Kernel Runtime for Generating 128 Tokens**
>
> | LLaMA Model | Baseline (FP16) | SqueezeLLM (4-bit) | SqueezeLLM (4-bit, 0.45% outliers) | SqueezeLLM (3-bit) | SqueezeLLM (3-bit, 0.45% outliers) |
> | ----------- | --------------- | ------------------ | ---------------------------------- | ------------------ | ---------------------------------- |
> | 7B          | 1.21            | 0.83               | 1.09                               | 0.56               | 0.83                               |
> | 13B         | 2.32            | 1.52               | 1.87                               | 0.97               | 1.32                               |
> | 30B         | 5.56            | 3.66               | 4.25                               | 2.26               | 2.86                               |
>
> ---
>
> ## W2. Feedback on Model Performance
>
>
> > Feedback on Model Performance: The majority of the model performance results focus on PPL outcomes. I believe it would be beneficial to include MMLU or CSR results for larger models within the main content, rather than relegating it to the appendix.
>
> Thank you for your constructive feedback. We acknowledge your point regarding the inclusion of MMLU and CSR results for larger models in the main body of the content. As can be seen in Table A.4 and A.7, our method *consistently outperforms* other methods both in terms of perplexity as well as MMLU accuracy on *larger models*, showing the strong transferability of the results in the smaller models to the larger models. The decision to place these results in the appendix was due to the page constraints, and we will include it in the main content in the final version. Reviewer 1 also asked us to perform 5-shot MMLU results which are included in response to their question (please see W3 of Reviewer U5Sb)

---

> > ### Author Response · Authors · 2023-11-22
> >
> > Dear Reviewer Ed31
> > \
> > \
> > We hope that our response has provided clarification to your concerns.
> > As today is the last day for author-reviewer communication, we would greatly appreciate it if you could please let us know if we could provide any further clarifications about the paper.
> >
> > Thanks so much again for taking the time to review our paper.
> > \
> > \
> > Best regards,
> >
> > SqueezeLLM Authors

---

> > > ### Comment · Reviewer_Ed31 · 2023-11-23
> > >
> > > Thank you for the detailed response. However, as I haven't reviewed the Revision paper, the points you've just raised are insufficient to elevate this paper to an acceptable level. Frankly, the issues I've identified are not easily resolved in a short period, and I acknowledge that substantial portions of the paper might need alteration. Including proper experimental and analytical results for the A100 Kernel instead of A6000, and shifting to more advanced evaluation methods like MMLU / CSR, will certainly demand considerable time.

---

> > > > ### Author Response · Authors · 2023-11-23
> > > >
> > > > We appreciate the follow up. We have indeed worked to address the reviewer’s questions.
> > > >
> > > > 1. We did include the requested benchmarks on A100 and added it to the response above (link:https://openreview.net/forum?id=pZhdz4oyzo&noteId=o9rHZ1PjFr). The results clearly show that our kernels do provide speed up on A100 as well and that our results are not limited to A6000 GPUs.
> > > >
> > > > 2. Furthermore, we also had included MMLU evaluation in our paper (please see Table 2). We furthermore, also included 5-shot MMLU evaluation as requested by the first reviewer (please see https://openreview.net/forum?id=pZhdz4oyzo&noteId=eZLXpG0kXz). Both results clearly show that SqueezeLLM performance improvement is consistent and is not just limited to perplexity improvement as compared to other methods.
> > > >
> > > > We would appreciate it if the reviewer could please let us know which specific questions have not been answered.

---

### Official Review · Reviewer_U5Sb · 2023-11-01

**Soundness:** 3 good
**Presentation:** 3 good
**Contribution:** 3 good
**Rating:** 5
**Confidence:** 3

**Summary:**

To address memory bandwidth constraints of large language models (LLMs), introduced SqueezeLLM, a post-training quantization framework, compresses LLMs to low precisions (to 4-bit or 3-bit range) without compromising performance. Two strategies underpin SqueezeLLM. One is sensitivity-based non-uniform quantization, which optimizes bit precision based on the weight distributions in LLMs. Second is dense-and-sparse decomposition, which stores outliers and simplifies the quantization process for the remaining weights. Extensive evaluations show that SqueezeLLM consistently surpasses existing quantization techniques across various bit precisions and tasks.

**Strengths:**

* The paper well explains the "memory wall" problem of LLMs and justifies the need for weight-only non-uniform quantization.
* The proposed method, SqueezeLLM, considers both the sensitivity and outlier of the weights while compressing using LLM's weight-only non-uniform quantization format.
* The proposed method demonstrates competitive performance across different sizes of models and tasks.
* Provide a latency report for the proposed kernel and compare it with others.
* The paper is well-structured and the proposed method is clearly elucidated.

**Weaknesses:**

* SqueezeLLM uses a small sample of the training dataset to execute end-to-end forward and backward passes for gradient computations. This process appears to be more resource-intensive than other methods like RTN, GPTQ, or AWQ. It would be beneficial to understand the time and resources required for SqueezeLLM's dense-and-sparse quantization.
* Relatedly, in the discussion about the need to minimize overall perturbations for the final loss term in section 4.1's "Sensitivity-Based K-means Clustering," the paper should also compare its performance with methods like AdaRound [1] or FlexRound [2]. These methods utilize a small calibration set and employ layer-wise or block-wise post-training quantization techniques. Notably, since FlexRound reports results on LLaMA using uniform weight-only quantization, it would strengthen the paper's claim about optimization with the final loss.
* It would be beneficial to display five-shot performance results on the MMLU benchmark using LLaMA, as this would offer a more comprehensive comparison with other methodologies.

[1]Nagel, Markus, et al. "Up or down? adaptive rounding for post-training quantization." International Conference on Machine Learning. PMLR, 2020.
[2]Lee, Jung Hyun, et al. "FlexRound: Learnable Rounding based on Element-wise Division for Post-Training Quantization." International Conference on Machine Learning. PMLR, 2023.

**Questions:**

* Is there a relationship between outliers and sensitive weights? Specifically, are most of the sensitive weights outliers?
* Have you experimented with combining a uniform quantization scheme with the sparse decomposition concept?
* In Table 1, there's a comparison with AWQ's latency performance. However, such a comparison is missing in Table 3 (section 5.4). Since the AWQ kernel is well showcased, it would be more convincing to show a comparison of the proposed kernel with AWQ.

---

> ### Author Response · Authors · 2023-11-16
> **Official Comment by Authors [1/2]**
>
> We appreciate the reviewer's constructive feedback, and here we address your comments in detail:
>
> ## W1. Resource Requirements
>
> > SqueezeLLM uses a small sample of the training dataset to execute end-to-end forward and backward passes for gradient computations. This process appears to be more resource-intensive than other methods like RTN, GPTQ, or AWQ. It would be beneficial to understand the time and resources required for SqueezeLLM's dense-and-sparse quantization.
>
> That is correct and SqueezeLLM involves an end-to-end forward and backward pass over a small set of sampled data. This method differs from other methods such as RTN, which does not require any forward or backward pass, and GPTQ or AWQ, which requires isolated gradient calculation over individual layers, resulting in lower overhead. However, it is important to note that the overhead of SqueezeLLM is quite manageable and, crucially, our method consistently outperforms RTN, GPTQ, and AWQ as shown in the paper (Table 1).
> To provide a clearer understanding of SqueezeLLM’s resource requirements, we conducted a comprehensive analysis of the time and memory requirements for (1) Fisher information computation and (2) sensitivity-aware k-means clustering for nonuniform quantization as below.
> \
> \
> **1. Fisher Information Computation:**
>
> | LLaMA Model | End-to-End Latency (s) | Peak Memory (GB) |
> | --- | -- | -- |
> | 7B   | 20     | 33   |
> | 13B   | 35     | 61               |
> | 30B         | 80   | 149              |
> | 65B         | 150     | 292              |
>
>
> We assessed the end-to-end latency and peak memory usage for computing Fisher information on an A100 system across different LLaMA models. The results indicate that the latency for calculating Fisher information (i.e. the sum of squares of gradients) is pretty fast and quantizing the 65B model can be performed on 4 A100 GPUs to address the peak memory usage.
> \
> \
> **2. Sensitivity Aware K-means Clustering:**
>
> | LLaMA Model | End-to-End Latency (min) |
> | -- | ----- |
> | 7B  | 18    |
> | 13B   | 33  |
> | 30B  | 71  |
> | 65B  | 120 |
>
>
> Another key component of SqueezeLLM is the sensitivity-aware K-means clustering applied to each weight matrix. This process can be executed efficiently in parallel across different layers, possibly using multiple cloud CPU instances. Our latency measurements on 8 Intel Xeon Platinum 8380 systems show that the time overhead for this clustering process is quite manageable.
> We will include these data in the final version of the paper to offer a comprehensive view of SqueezeLLM's time and resource demands.
>
> ---
>
> ## W2. Comparison with FlexRound
>
> > Relatedly, in the discussion about the need to minimize overall perturbations for the final loss term in section 4.1's "Sensitivity-Based K-means Clustering," the paper should also compare its performance with methods like AdaRound [1] or FlexRound [2]. These methods utilize a small calibration set and employ layer-wise or block-wise post-training quantization techniques. Notably, since FlexRound reports results on LLaMA using uniform weight-only quantization, it would strengthen the paper's claim about optimization with the final loss.
>
>
> That is a great question. Below in the table, we provide a comparison with AdaRound and FlexRound. We first have to kindly emphasize that it is unfortunately not feasible to do a direct comparison with FlexRound due to their closed-source code and models, as well as a lack of experimental details for their perplexity evaluation on WikiText. However, to address the reviewer’s question to the best of our ability,  we have utilized the results published in the FlexRound paper (for AdaRound and FlexRound) to assess the amount of post-quantization perplexity degradation of each method across different LLaMA models. As one can see, SqueezeLLM consistently exhibits a smaller drop in perplexity compared to AdaRound and FlexRound.
>
> We should also kindly emphasize that we have performed a comprehensive comparison with the *latest weight-only quantization schemes* designed for LLMs, such as AWQ and SpQR, which share the same goal of minimizing output activation perturbation as FlexRound. In these comparisons, our method consistently demonstrates superior performance, both with 4-bit and 3-bit. Therefore, we maintain that our sensitivity-based k-means clustering that optimizes the final loss provides a more effective post-training quantization solution compared to other existing quantization strategies.
> \
> \
> **Perplexity on Wikitext**
>
> | LLaMA Model | AdaRound Perplexity Drop (baseline →  quantized) | FlexRound Perplexity Drop (baseline →  quantized) | SqueezeLLM Perplexity Drop (baseline →  quantized) |
> | ----------- | -- | -- | --- |
> | 7B          | 0.79 (8.90 → 9.69)        | 0.28 (8.90 → 9.18)    | **0.09 (5.68 → 5.77)**     |
> | 13B         | 0.34 (7.73 → 8.07)    | 0.16 (7.73 → 7.90)   | **0.09 (5.09 → 5.18)**     |
> | 30B         | 0.53 (6.35 → 6.88)     | 0.28 (6.35 → 6.63)     | **0.08 (4.10 → 4.18)**   |

---

> ### Author Response · Authors · 2023-11-16
> **Official Comment by Authors [2/2]**
>
> ## W3. 5-shot MMLU Performance
>
>
> > It would be beneficial to display five-shot performance results on the MMLU benchmark using LLaMA, as this would offer a more comprehensive comparison with other methodologies.
>
> That is an excellent question. We have included five-shot performance results on the MMLU benchmark for Vicuna-v1.1 in the table below and also included a comparison with AWQ with the same model size. Similar to the zero-shot results that we had included in the paper, we can see that SqueezeLLM **consistently achieves higher-quality results**. At the moment, we are also running a comparison with GPTQ but we expect GPTQ to have a similar performance as AWQ. We will include these results in the final version of the paper beside Table 2 where we compared instruction following capabilities of the different models.
> \
> \
> **MMLU 5-shot Accuracy**
>
> | Vicuna-v1.1 | Baseline (FP16) | SqueezeLLM (3.24 bit) | AWQ (3.24 bit) |
> | --| -- | -- | --------------------------------------- |
> | 7B          | 45.3%  | **42.2%**  | 41.4%|
> | 13B         | 50.0%  | **48.2%**   | 46.3% |
>
>
> ---
>
>
> ## Q1. Outliers vs. Sensitive Values
>
>
> > Is there a relationship between outliers and sensitive weights? Specifically, are most of the sensitive weights outliers?
>
> From our extensive experiments conducted on Vicuna, LLaMA, and LLaMA2 models, we found no direct correlation between outliers and sensitive weights. In fact, we frequently observed that weights with smaller values can also be sensitive (as in Figure 2, Left). This is expected since the sensitivity measured by the Fisher information is not necessarily correlated with the magnitude of the weight value, but more with the curvature of the loss function with respect to that weight value.
>
> ---
>
> ## Q2. Uniform Quantization + Dense-and-Sparse Decomposition
>
>
> > Have you experimented with combining a uniform quantization scheme with the sparse decomposition concept?
>
> Yes, we have indeed explored integrating uniform quantization with the dense-and-sparse decomposition while developing SqueezeLLM. Specifically, we applied dense-and-sparse decomposition in combination with channel-wise uniform quantization on LLaMA 7B and 13B models. In the table below, we compared the performance of uniform quantization and our sensitivity-aware non-uniform quantization, both paired with the dense-and-sparse decomposition. The results indicate that non-uniform quantization based on our sensitivity-aware K-means method yields significantly better perplexity.
> It is worth noting that the performance of uniform quantization could potentially be improved through grouped quantization strategies, where a small set of weights (e.g. 128 as in the table below) are grouped together to have their own scaling factor. While grouped quantization, as demonstrated in techniques like GPTQ and AWQ, is a viable approach, the SqueezeLLM approach offers a much simpler implementation by avoiding the complexities of fine-grained grouping and it still outperforms uniform quantization methods with grouping.
> \
> \
> **Perplexity on C4**
>
> | LLaMA Model | Uniform | Dense-and-Sparse + Uniform | Dense-and-Sparse + Uniform with grouping = 128 | Dense-and-Sparse + Non-uniform |
> | ----------- | ------- | -------------------------- | ---------------------------------------------- | ------------------------------ |
> | 7B          | 28.26   | 9.25                       | 8.32                                           | **7.56**                       |
> | 13B         | 13.24   | 7.96                       | 7.30                                           | **6.92**                       |
>
> ---
>
> ## Q3. Comparison with AWQ kernels
>
>
> > In Table 1, there's a comparison with AWQ's latency performance. However, such a comparison is missing in Table 3 (section 5.4). Since the AWQ kernel is well showcased, it would be more convincing to show a comparison of the proposed kernel with AWQ.
>
> We appreciate the suggestion to include a comparison with AWQ's public kernel. However, as we outlined in Appendix A.3, we chose not to compare our kernels directly with those from AWQ for a couple of key reasons: (1) The 3-bit kernels of AWQ are not publicly available; and (2) AWQ has incorporated optimizations that are unrelated to quantization, such as LayerNorm and positional embedding. These optimizations, while effective, are universally applicable to various quantization methods including SqueezeLLM and GPTQ.

---

> ### Comment · Reviewer_U5Sb · 2023-11-21
>
> Dear Author,
> Thank you for responding to my previous concerns.
> I have an additional question regarding Equation (4) in your paper. It appears that Equation (4) requires a more explicit derivation. The interpretation of Equation (4) as the minimization problem involving the sum of the Fisher Information for the weight difference between the original and quantized weights is somewhat unclear to me. Specifically, I am struggling to understand the connection between this concept and “minimizing the overall perturbation with respect to the final loss term.” which is the term on the paper.
> To the best of my knowledge, as demonstrated in the BRECQ paper [1], the derivation of Equation (3) should conclude with an expression similar to:
>
> $argmin$ $\sum_{i=1}^{N}$ $\mathscr{F}$ $(y-\hat{y})^2$
>
> where y is an original output of each layer, $\hat{y}$ is an output of the quantized layer.
> For additional context, I refer you to Appendix A.1, “Proof of Theorem 3.1,” in the BRECQ paper on page 12 [1].
>
> [1] https://openreview.net/pdf?id=POWv6hDd9XH

---

> > ### Author Response · Authors · 2023-11-22
> >
> > Thank you for reviewing our response and for your question.
> >
> > In fact, the BRECQ paper also starts with the same goal of minimizing the perturbation of the final loss term.
> > In particular, please note that Equation 4 in the BRECQ paper [1] is the same as Equation 3 in our paper.
> >
> > However, the BRECQ paper then applies an *approximation* in Theorem 3.1 of their paper, approximating
> > $argmin_\theta  \Delta \theta H^{(\theta)} \Delta \theta$ (Equation 4) with  $argmin_{\theta} E [\Delta z^T H^{(z)}\Delta z]$ (Equation 7) where $\theta$ is the model weight and $z$ is the output activation.
> > Then they proceed with a further approximation using the Fisher information *with respect to activations*, which leads to $\text{argmin} \sum_{i=1}^N \mathcal{F}(z - \hat{z})^2$ as what the reviewer mentioned.
> >
> > Please note that, first, the derivation in the BRECQ paper involves an *additional approximation process*, as highlighted in their Theorem 3.1, which transforms Equation 4 in their paper to Equation 7.
> > Second, their approach requires the Fisher information *with respect to activations*, whereas our method uses the Fisher information *with respect to weights*
> >
> > In summary, our approach **directly computes the Fisher information with respect to weights** as a means to approximate the Hessian in Equation 3 of our paper, which leads us to the optimization objective presented in Equation 4.
> >
> > [1] Li, Yuhang, et al. "Brecq: Pushing the limit of post-training quantization by block reconstruction." arXiv preprint arXiv:2102.05426 (2021).

---

> > > ### Author Response · Authors · 2023-11-22
> > >
> > > Dear Reviewer U5Sb
> > > \
> > > \
> > > We hope that our response has provided clarification to your concerns.
> > > As today is the last day for author-reviewer communication, we would greatly appreciate it if you could please let us know if we could provide any further clarifications about the paper.
> > >
> > > Thanks so much again for taking the time to review our paper.
> > > \
> > > \
> > > Best regards,
> > >
> > > SqueezeLLM Authors

---

> > > > ### Comment · Reviewer_U5Sb · 2023-11-23
> > > >
> > > > Dear Author,
> > > >
> > > > Thank you for your prompt and informative response regarding the derivation process in your paper.
> > > >
> > > > While I appreciate the detailed explanation responding to my concern, I find it challenging to clearly discern why differentiating with respect to weights is more advantageous than differentiating with respect to outputs, as more commonly seen in the literature. Additionally, a clear derivation showing how this approach leads to the optimization objective presented in Equation 4 is essential to validate your methodology.
> > > >
> > > > Therefore, I kindly request that you provide a more explicit and detailed derivation of this process in your paper.

---

> > > > > ### Author Response · Authors · 2023-11-23
> > > > >
> > > > > Thank you so much for your quick follow up. Below we provide more details which we hope to address the reviewer’s concern.
> > > > >
> > > > > The derivation of Equation 4 is pretty standard and used in previous papers, notably [1]. Quantization is being performed to the weights and not activations. We would like to find a quantization scheme that results in minimal perturbation to the output of the model. That is why we start with Equation 2 of our paper and perform a Taylor series expansion for the perturbation to the weights. Expanding this results in Equation 4. This is basically a standard Taylor series expansion at the optimum point. Please note that this is also the same as Equation 4 of BRECQ paper [2].
> > > > >
> > > > > We are not sure which part of this has caused confusion and we would appreciate it if the reviewer could please let us know so we can clarify it. This equation is used in both [1,2] as well as our paper, which is technically Taylor series expansion at the optimum point.
> > > > >
> > > > > Now the work of BRECQ uses an **approximation** in Theorem 3.1 to show that under certain assumptions using the Hessian wrt activations is equivalent to Equation4 of their paper. In particular, you can see the source of this approximation in their paper in Equation 8 of their work where they use Finite differences to approximate gradient wrt weights with perturbation to the activations. This assumption is only valid if the perturbations are small. They then proceed with this approximation and use the second derivative wrt activations, approximate it with Fisher information wrt activations, and then arrive at Equation 10. Furthermore, BRECQ uses this approach along with QAT (please see Algorithm 1 of their paper). We are not performing any QAT here.
> > > > >
> > > > > We should also emphasize that the approximation used in BRECQ mentioned above is not needed in our method. We directly have access to the Fisher information wrt weights, so we can directly use that information.
> > > > >
> > > > > Also it is worth noting that we need to use the formation wrt weights because we need to calculate the non-uniform quantization by performing a sensitivity based k-means for the **weights**. Using the Fisher information wrt weights results in a very straightforward calculation for the non-uniform quantization as Equation 4 can be directly fed into a k-means algorithm to find the best set of non-uniform **weights** to represent the distribution in quantized precision.
> > > > >
> > > > > Finally, we should emphasize that some quantization approaches such as GPTQ try to minimize the perturbation not to the end layer, but to the specific layer that is being quantized. This is basically following the Optimal Brain Surgeon method [4]. There has been several studies showing the different tradeoffs between Optimal Brain Damage [1] (which minimizes perturbations to the end layer/loss and is also used by our method) vs Optimal Brain Surgeon [4] (which minimizes perturbations to a specific layer in isolation and used by methods such as GPTQ). Our results in this paper shows a clear accuracy advantage of SqueezeLLM. This is expected since perturbation errors in a particular layer may not be important if the predictions of the network are not sensitive to that particular layer. We also included a dedicated section in our paper which makes this comparison (please see A.5.4 in our Appendix).
> > > > >
> > > > > We hope that this addresses the reviewer’s concerns.
> > > > >
> > > > >
> > > > >
> > > > > [1] Optimal brain damage. Yann LeCun, John Denker, and Sara Solla. NeurIPS 1989.
> > > > >
> > > > > [2] BRECQ: Pushing the limit of post-training quantization by block reconstruction. Yuhang Li et al. ICLR 2021
> > > > >
> > > > > [3] A Fast Post-Training Pruning Framework for Transformers. Woosuk Kwon et al. NeurIPS 2022
> > > > >
> > > > > [4] Optimal Brain Surgeon: Extensions and performance comparisons. Babak Hassibi et al. NeurIPS 1993.

---

> ### Comment · Reviewer_U5Sb · 2023-11-23
>
> I understand Eq. (2) in your paper that is the same as Eq. (4) in the BRECQ paper. In particular, it is worth noting that $H$ in Eq. (2) in your paper is exactly the same as $\bar{\text{H}}^{(w)}$ in Eq. (4) in the BRECQ paper.
>
> In the BRECQ paper, $\Delta w \bar{\text{H}}^{(w)} \Delta w$ is approximated to $\Delta z \bar{\text{H}}^{(z)} \Delta z$ by using Theorem 3.1 in the BRECQ paper, where  $\bar{\text{H}}^{(z)}$ is the output Hessian that is totally different from $\bar{\text{H}}^{(w)}$. Then, the output Hessian $\bar{\text{H}}^{(z)}$ is approximated to the Fisher Information Matrix $\mathcal{F}$ as shown in Eq. (10) in the BRECQ paper.
>
> On the contrary, in your paper, $H$ in both Eq. (2) and Eq. (3) is essentially $\bar{\text{H}}^{(w)}$, but then $\bar{\text{H}}^{(w)}$ is approximated to the Fisher Information Matrix $\mathcal{F}$ in Eq. (4) without any explanation. So, I would like to point out how the authors can derive the Fisher Information Matrix $\mathcal{F}$ from $\bar{\text{H}}^{(w)}$.
>
> The authors mentiond [1], but to the best of my knowledge, like the BRECQ paper, [1] also handled the pre-activation Hessian $\partial^2E / \partial a_i^2$ as seen in Eq. (7) and (8) in [1].
>
> [1] Optimal brain damage. Yann LeCun, John Denker, and Sara Solla. NeurIPS 1989.

---

### Meta-Review · Area_Chair_ogzH · 2023-12-10

**Metareview:**

This paper proposes a post training weight quantization method for LLMs, that pushes the average numerical precision to 3-4 bits. While reviewers agree that this is a well-written paper with competitive results, some consideration remains:
- more thorough wall clock time results
- confusion approximation of Eq. (4)
- lack of discussion on some critical works in related work that have introduced similar techniques.
Authors do partially address these concerns in their feedbacks. However, the web-based responses do not fully convince the reviewers, and there lacks a revised version of the paper. All reviewers agree that the paper cannot be accepted in its current form.

**Justification For Why Not Higher Score:**

The authors' responses do not fully convince the reviewers of the considerations, and there lacks a revised version of the paper.

**Justification For Why Not Lower Score:**

N/A

---

### Decision · Program_Chairs · 2024-01-16

Reject